# Tyrosine phosphorylation regulates RIPK1 activity to limit cell death and inflammation

Hailin Tu[1], Weihang Xiong[1,2], Jie Zhang[1], Xueqiang Zhao[1] & Xin Lin [1,2] ✉

Receptor-interacting serine/threonine-protein kinase 1 (RIPK1) is a cytosolic protein kinase that regulates multiple inflammatory and cell death pathways. Serine/Threonine phosphorylation of RIPK1 is known to suppress RIPK1 kinase-mediated cell death in the contexts of inflammation, infection and embryogenesis, however, regulation by tyrosine phosphorylation has not been reported. Here, we show that non-receptor tyrosine kinases Janus kinase 1 (JAK1) and SRC are able to phosphorylate RIPK1 at Y384 (Y383 in murine RIPK1), leading to suppression of TNF-induced cell death. Mice bearing a homozygous Ripk1 mutation that prevents tyrosine phosphorylation of RIPK1 (Ripk1Y383F/Y383F), develop systemic inflammation and emergency haematopoiesis. Mechanistically, Ripk1Y383F/Y383F mutation promotes RIPK1 kinase activation and enhances TNF-induced apoptosis and necroptosis, which is partially due to impaired recruitment and activation of MAP kinase-activated protein kinase 2 (MK2). The systemic inflammation and emergency haematopoiesis in Ripk1Y383F/Y383F mice are largely alleviated by RIPK1 kinase inhibition, and prevented by genomic deletions targeted to the upstream pathway (either to Tumor necrosis factor receptor 1 or RIPK3 and Caspase8 simultaneously). In summary, our results demonstrate that tyrosine phosphorylation of RIPK1 is critical for regulating RIPK1 activity to limit cell death and inflammation.

RIPK1 (Receptor interacting serine/threonine-protein kinase 1) is essential for regulating cell death and inflammation signaling pathway[1]. The function of RIPK1 in cell fate determination is well studied in tumor necrosis factor receptor 1 (TNFR1) signaling pathway[2]. Upon TNF ligation of TNFR1, signaling molecules including TNFR1-associated death domain protein (TRADD), RIPK1, TNF receptor associated factor 2/5 (TRAF2/5), Cellular inhibitor of apoptosis protein-1/2 (cIAP1/2) and Linear ubiquitin chain assembly complex (LUBAC) are rapidly recruited to generate ubiquitin chains on RIPK1, which could serve as a scaffold to recruit downstream IκB kinase α/β (IKKα/β)-NEMO and tumor growth factor-β-activated kinase 1 (TAK1)-TAB2/3 complex. These signaling components assemble to form TNFR1 signaling complex (TNF-RSC, also called complex I), further activating downstream mitogen-activated protein kinase (MAPK) and

nuclear factor-κB (NF-κB) signaling for cell survival[3,4]. Nevertheless, RIPK1 and/or TRADD could associate with Fas-associated death domain protein (FADD) and Caspase8 in the cytoplasm to form complex II when the formation of TNF-RSC is disrupted, which consequently induces apoptosis[5,6]. In addition, when apoptosis is inhibited by the inactivation of Caspase8, RIPK1 in complex II promotes activation of Receptor-interacting serine/threonine-protein kinase 3 (RIPK3) and Mixed lineage kinase domain-like protein (MLKL) in a kinase-dependent way to trigger necroptosis execution[7–9].

The scaffold and kinase activity of RIPK1 in TNFR1 signaling is tightly controlled by post-translational modifications, especially by phosphorylation, ubiquitination, and Caspase-mediated cleavage[10]. Ubiquitination of RIPK1 is crucial for its scaffold activity to promote NF-κB activation and limiting its kinase activity to suppress RIPK1-

[1]Institute for Immunology, School of Medicine, Tsinghua University, Beijing, China. [2]Tsinghua University–Peking University Center for Life Sciences, Beijing 100084, China. ✉e-mail: linxin307@tsinghua.edu.cn

dependent cell death during embryogenesis and inflammation[11–14]. In addition, genetic and clinical studies have revealed a critical role of cleavage of RIPK1 by Caspase8 in preventing RIPK1 kinase-mediated apoptosis and necroptosis[15–18]. Of note, numerous recent studies have reported phosphorylation of RIPK1 mediated by TAK1, MAP kinase-activated protein kinase 2 (MK2), TBK1 (TANK binding kinase 1)/IKKε and IKKα/β provide a physiological brake to prevent TNF-induced RIPK1 kinase-dependent cell death during embryogenesis, infection, and neuroinflammation[19–26]. These phosphorylation modifications are mainly limited to serine or threonine of RIPK1. Although hyperactivation of tyrosine phosphorylation of RIPK1 by receptor tyrosine kinase MET was reported to promote colon cancer progression, the physiological function and molecular mechanism how tyrosine phosphorylation regulates RIPK1 activity in TNFR1 signaling remains poorly unknown[27]. Thus, uncovering the role of tyrosine phosphorylation on RIPK1 will deepen our understanding on the phosphorylation-mediated regulation of RIPK1 activity in TNFR1 signaling.

Non-receptor tyrosine kinase SRC has been reported to phosphorylate Caspase8 to regulate TNF-induced apoptosis[28], suggesting tyrosine phosphorylation could participate in regulation of cell death signaling. In addition, non-receptor tyrosine kinase JAK1 (Janus Kinase 1), a key regulator in interferon signaling, is well-studied for playing a critical role in immunomodulatory and antiviral functions[29]. Previously, several studies have shown that interferon signaling components such as JAK1, JAK2, and STAT1 could participate in TNF-mediated JAK/STAT signaling activation through TNFR1[30,31]. However, whether JAK-STAT signaling components could modulate TNFR1-induced cell death especially RIPK1 activity remains unknown.

In this study, we show that non-receptor tyrosine kinases JAK1 and SRC could phosphorylate RIPK1 on tyrosine 384 (Y384) in human RIPK1 (Y383 in mouse RIPK1), and serve as essential regulators of RIPK1 in the TNFR1 signaling pathway. We determine the physiological function of tyrosine phosphorylation of RIPK1 via generating *Ripk1^{Y383F/Y383F}* mice. *Ripk1^{Y383F/Y383F}* mice develop systemic inflammation and sustained emergency hematopoiesis due to severe TNF-induced RIPK1 kinase-dependent apoptosis and necroptosis. We elucidate the mechanism that tyrosine phosphorylation of RIPK1 on Y383 limits RIPK1 kinase-dependent cell death partially through enabling recruitment and activation of MK2. Together, our studies provide genetic and molecular evidence how tyrosine phosphorylation of RIPK1 on Y383 orchestrates RIPK1-dependent cell death and further regulates inflammation.

## Results

### Identification of tyrosine phosphorylation as potential regulation of RIPK1 activity

To elucidate whether signaling components that might participate in the regulation of RIPK1 activity, we over-expressed Flag-tagged RIPK1 in HEK293T cells and potential RIPK1-associated proteins were determined via mass spectrometry analysis (Supplementary Fig. 1a). As expected, TNF-RSC components such as TRAFs, cIAP1, MIB1/2, TAB2/3 and LUBAC, and cell death complex components such as FADD, Caspase8, TRADD, and cFLIP were co-purified together with RIPK1 (Table 1 and Supplementary Table 1). In addition, kinase complexes such as TAK1, TAB2, MK2, and NEMO were also strongly associated with RIPK1 (Table 1). Notably, non-receptor tyrosine kinases JAK1 and SRC were also co-purified with RIPK1 (Table 1). We next verified the interaction of JAK1 with RIPK1 via co-immunoprecipitation and found that JAK1 and SRC indeed have a strong interaction with RIPK1 (Fig. 1a, b and Supplementary Fig. 1b). However, we did not observe SRC could interact with JAK1 with RIPK1 existence (Supplementary Fig. 1c), suggesting the two non-receptor tyrosine kinases function independently to regulate RIPK activity. To investigate whether JAK1 and SRC could regulate RIPK1 in TNF-TNFR1 signaling, we surprisingly found that JAK1 but not SRC could be recruited to TNF-RSC upon TNFα stimulation (Fig. 1c).

## Table 1 | Identification of RIPK1 interacting proteins via mass spectrometry analysis

| RIPK1-mass spectrometry | | Protein | PSM |
|---|---|---|---|
| E3 ligase complex (Ubiquitination) | | TRAF2 | 32 |
| | | cIAP1 | 23 |
| | | MIB1 | 16 |
| | | HOIP | 6 |
| | | SHARPIN | 1 |
| | | HOIL1 | 1 |
| | | TNIP1 | 6 |
| Kinase complex (Phosphorylation) | | TAB2 | 24 |
| | | TAK1 | 15 |
| | | NEMO | 6 |
| | | MK2 | 2 |
| Unknown | | **JAK1** | 2 |
| | | **SRC** | 5 |

Consistently, JAK1 could endogenously interact with RIPK1 under TNFα stimulation alone (Fig. 1d). Moreover, we found SRC but not JAK1 could interact with RIPK1 in TNFR1 complex II under TNF and Smac mimetics (BV-6) stimulation (Fig. 1e). These results suggest that JAK1 and SRC could involve in TNFR1 signaling to regulate RIPK1 activity.

JAK1 and SRC has been well-known as essential tyrosine kinases to phosphorylate substrates in various cellular signaling pathway including interferon and cell death signaling[28,32,33]. Recent studies have also demonstrated the crucial role of phosphorylation of RIPK1 in suppressing RIPK1 enzymatic activity to modulate cell death signaling[19–26]. Thus, we hypothesized that JAK1 and SRC could also act as a cytosolic tyrosine kinase to phosphorylate RIPK1 during TNFR1 signaling. We transfected expression vectors of Flag-RIPK1 and HA-JAK1/SRC in HEK293T cells and found RIPK1 could be tyrosine phosphorylated when co-expressed with JAK1 or SRC (Fig. 1f, g). In addition, we treated WT primary BMDMs (Bone marrow-derived macrophages) with TNF or TNF-BV-6 and found RIPK1 could indeed undergo tyrosine phosphorylation endogenously in TNFR1 signaling (Fig. 1h). Consistently, pretreatment with JAK1 or SRC inhibitor could dramatically block tyrosine phosphorylation on RIPK1 (Fig. 1h), suggesting that tyrosine phosphorylation on RIPK1 depends on JAK1 or SRC kinase. Taken together, these results suggest that tyrosine phosphorylation could be an essential regulation of RIPK1 in TNFR1 signaling.

### Y384 is a major tyrosine phosphorylation site of RIPK1

To better understand the function of tyrosine phosphorylation of RIPK1, we sought to identify the specific tyrosine phosphorylation site of RIPK1. Similarly, we expressed different truncation vectors of RIPK1 with JAK1 and found that intermediate domain of RIPK1 is responsible for interaction with JAK1 (Fig. 2a, b and Supplementary Fig. 2a, b). In addition, JAK1 could trigger tyrosine phosphorylation in intermediate domain of RIPK1 alone, suggesting that the tyrosine phosphorylation site of RIPK1 by JAK1 is located in intermediate domain (Fig. 2c). Furthermore, by searching PhosphoSitePlus database[34] (https://www.phosphosite.org), we found that RIPK1 has several unchartered tyrosine phosphorylation sites in intermediate domain—Y384, Y387, Y463, Y469, Y531, and Y534—with relative higher high-throughput references (Supplementary Fig. 2c). We then generated tyrosine (Y) to Phenylalanine (F) mutants of RIPK1 and found that only the Y384F/Y387F mutant could largely block tyrosine phosphorylation of RIPK1 by JAK1 (Fig. 2d). By analyzing the primary sequence of RIPK1 from different species, we found that only Y384 in human RIPK1 is an evolutionarily conserved residue in murine RIPK1 as Y383 (Fig. 2e). Accordingly, the Y384F

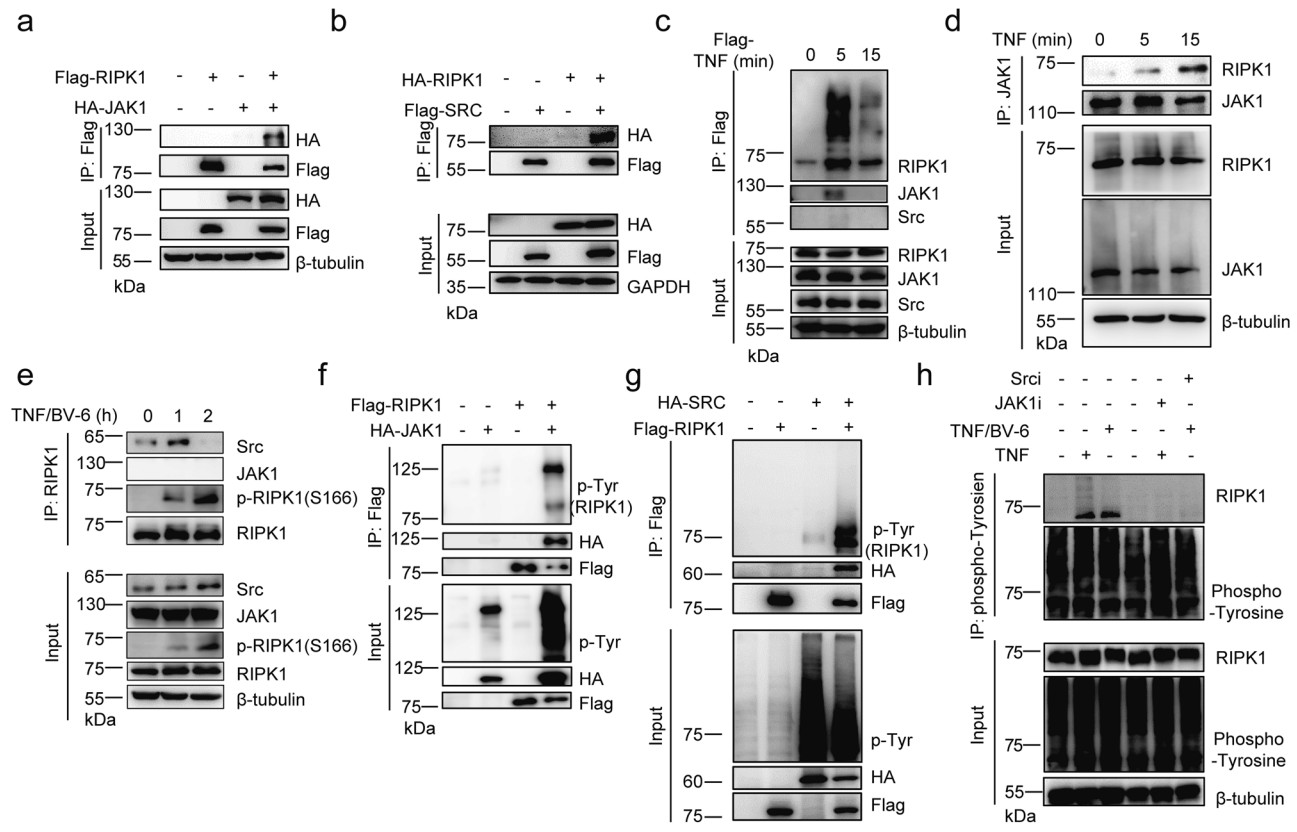

**Fig. 1 | Identification of tyrosine phosphorylation as potential regulation of RIPK1 activity.** HEK293T cells were co-transfected with expression vectors for Flag-RIPK1 and HA-JAK1 (**a**) or Flag-SRC and HA-RIPK1 (**b**) for 24 h. Cell lysates were then immunoprecipitated with anti-Flag-Protein A/G agarose and analyzed by immunoblotting with the indicated antibodies. **c** Primary WT BMDMs were stimulated by Flag-TNF (100 ng/ml) at indicated time points and TNF-RSC was immunoprecipitated by anti-Flag resin for western blotting with indicated antibodies. Primary WT BMDMs were stimulated by TNF (**d**) or TNF/BV-6 (**e**) at indicated time points and whole-cell lysates were immunoprecipitated using anti-JAK1 (**d**) or anti-

RIPK1 (**e**) antibody for immunoblotting with the indicated antibodies. HEK293T cells were co-transfected with expression vectors for Flag-RIPK1 and HA-JAK1 (**f**) or HA-SRC (**g**) for 24 h. Cell lysates were immunoprecipitated with anti-Flag-Protein A/G agarose analyzed by immunoblotting with phosphor-tyrosine antibody and other antibodies. **h** Primary WT BMDMs were stimulated by TNF for 15 min or TNF/BV-6 for 1 h with or without JAK1 and SRC inhibitor treatment. Cell lysates were then harvested and immunoprecipitated using anti-phospho-Tyrosine antibody for immunoblotting with the indicated antibodies. TNF: 10 ng/ml; BV-6: 2.5 uM. Source data are provided with this paper.

mutant of RIPK1 alone could greatly block tyrosine phosphorylation of RIPK1 by JAK1 or SRC (Fig. 2f, g). Thus, we conclude that Y384 in human RIPK1 (Y383 in murine RIPK1) is a major site for tyrosine phosphorylation of RIPK1.

## Loss of tyrosine phosphorylation of RIPK1 promotes TNF-induced apoptosis and necroptosis

To further investigate the role of tyrosine phosphorylation of RIPK1 in TNF-induced cell death signaling, we pre-treated BMDMs and MEFs with JAK1 or SRC kinase inhibitor to test their response to TNF-induced toxicity. Strikingly, pre-treatment with JAK1 or SRC inhibitor greatly sensitized BMDMs and MEFs to TNF-induced cell death, which could be inhibited by treatment with RIPK1 kinase inhibitor Nec-1 (Necrostatin-1) (Fig. 3a and Supplementary Fig. 3a–c). Interestingly, we found WT BMDMs co-treated with JAK and SRC inhibitor under TNF or TNF/BV6 stimulation could trigger higher level of cell death compared to treatment with JAK or SRC inhibitor alone (Fig. 3a and Supplementary Fig. 3a). These results suggest that JAK and SRC has a non-redundant function in protection from TNF-induced cell death. We next sought to investigate the role of tyrosine phosphorylation of RIPK1 on Y383 in TNFR1 signaling. We generated $Ripk1^{Y383F/Y383F}$ knock-in mice and found $Ripk1^{Y383F/Y383F}$ mutation in BMDMs and MEFs could greatly enhanced TNF-induced cell death in a RIPK1-dependent manner

(Fig. 3b, c and Supplementary Fig. 3d–f). $Ripk1^{+/+}$ and $Ripk1^{Y383F/Y383F}$ BMDMs pretreated with JAK1 and SRC inhibitor showed similar sensitivity to TNF-induced cell death, suggesting that tyrosine phosphorylation of RIPK1 on Y383 accounts for the protective role of JAK1 and SRC from TNF-induced cell death (Fig. 3b). In addition, we also treated BMDMs and MEFs with TNF plus SMAC mimetic BV-6 and found that $Ripk1^{Y383F/Y383F}$ mutation could enhance Caspase3 activation and RIPK1 kinase-dependent apoptosis (Fig. 3d, e and Supplementary Fig. 3g, h). Pre-treatment with pan-caspase inhibitor zVAD.fmk triggered increased phosphorylated MLKL and downstream necroptosis-mediated cell death in $Ripk1^{Y383F/Y383F}$ BMDMs and MEFs, suggesting that $Ripk1^{Y383F/Y383F}$ mutation also promotes TNF-induced necroptosis activation (Fig. 3d, f and Supplementary Fig. 3g, i).

We next sought to investigate the cell death machinery that contributes to enhanced TNF-induced cell death in $Ripk1^{Y383F/Y383F}$ MEFs. RIPK1 has been demonstrated to participate in Complex IIa and IIb formation to trigger apoptosis and necroptosis induction[2]. We, therefore, hypothesized tyrosine phosphorylation of RIPK1 on Y383 could limit cell death complexes formation to suppress apoptosis and necroptosis. As expected, $Ripk1^{Y383F/Y383F}$ mutation promotes RIPK1 activation and recruitment of RIPK1 associated cleaved Caspase-8, cleaved cFLIP, and FADD in Complex IIa upon TNF/BV6-induced RIPK1-dependent apoptosis (Fig. 3g). Furthermore, increased RIPK1

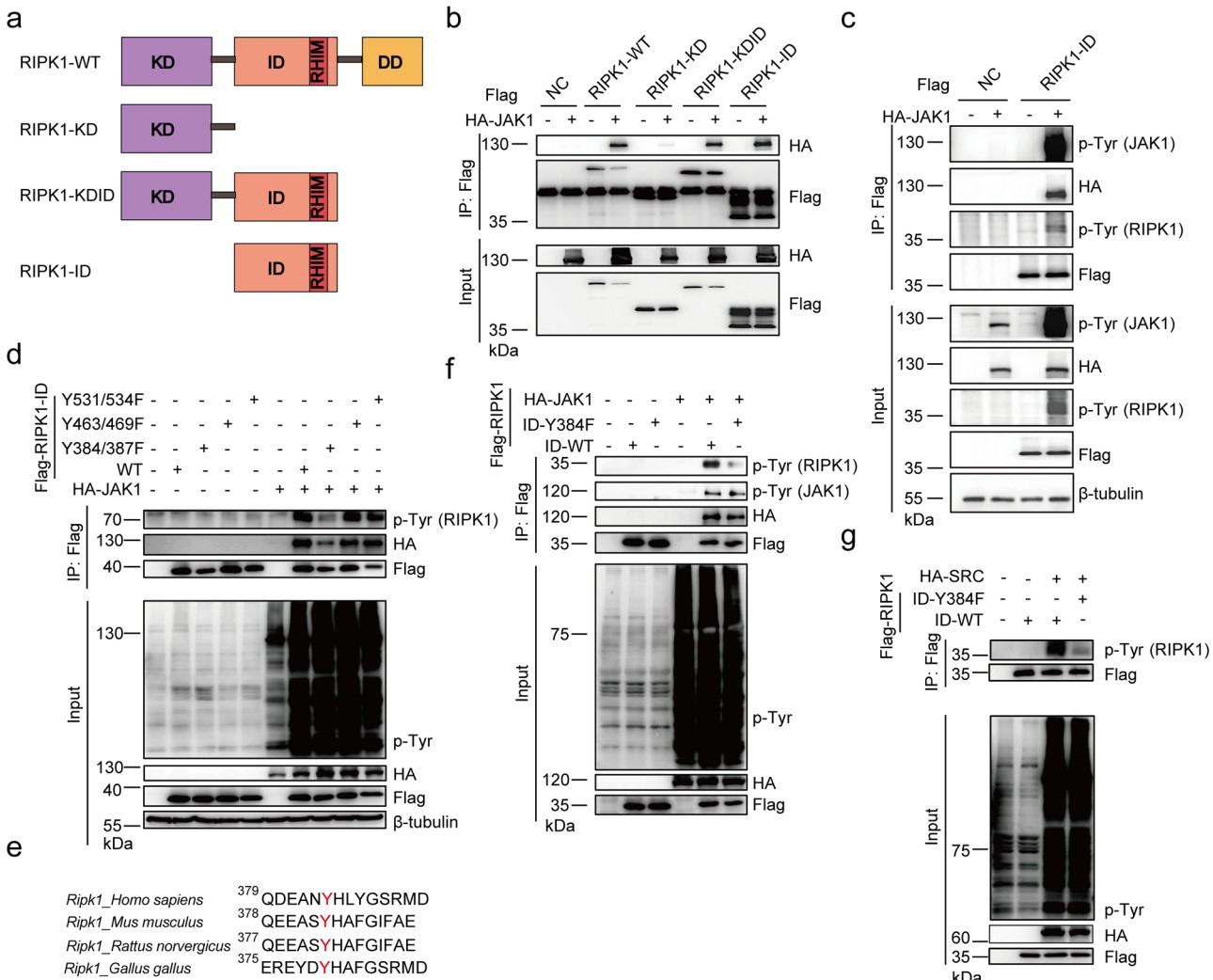

**Fig. 2 | Y384 is a major tyrosine phosphorylation site of RIPK1. a** Schematic overview of domain structure of different RIPK1 mutant plasmids: RIPK1-WT, RIPK1-KD (Kinase Domain), RIPK1-KDID (C-terminal Death Domain deletion), and RIPK1-ID (Intermediate Domain). HEK293T cells co-transfected with Flag-RIPK1-WT/KD/KDID/ID (**b**) or Flag-RIPK1-ID (**c**) and HA-JAK1 for 24 h. Cell lysates were immunoprecipitated with anti-Flag-Protein A/G agarose and analyzed by immunoblotting with the indicated antibodies. **d** Cell lysates of HEK293T cells co-transfected with Flag-RIPK1-WT, Y384F/387F, Y463/Y469F/Y531F/Y534F and HA-JAK1 for 24 h were immunoprecipitated with anti-Flag-Protein A/G agarose and analyzed by immunoblotting with the indicated antibodies. **e** Sequence alignment of human RIPK1 (*Homo sapiens*), murine RIPK1 (*Mus musculus*), rat RIPK1 (*Rattus norvegicus*), and chicken RIPK1 (*Gallus gallus*) using Clustal Omega is shown around Y384 and Y387. Cell lysates of HEK293T cells co-transfected with Flag-RIPK1-ID-WT/Y384F and HA-JAK1 (**f**) or HA-SRC (**g**) for 24 h were immunoprecipitated with anti-Flag-Protein A/G agarose and analyzed by immunoblotting with the indicated antibodies. Source data are provided with this paper.

activation and formation of necrosome comprised of RIPK1 and RIPK3 were observed in *Ripk1^{Y383F/Y383F}* MEFs upon TNF/BV-6/zVAD stimulation (Fig. 3h). Collectively, these results suggest that *Ripk1^{Y383F/Y383F}* mutation promotes cell death complexes formation to induce RIPK1-dependent apoptosis and necroptosis

**Tyrosine phosphorylation is critical for limiting RIPK1 kinase activity**

To determine whether *Ripk1^{Y383F/Y383F}* mutation impairs tyrosine phosphorylation of RIPK1 in endogenous level, we treated *Ripk1^{+/+}* and *Ripk1^{Y383F/Y383F}* primary MEFs and BMDMs with TNF and isolated tyrosine phosphorylated proteins via immunoprecipitation. Upon TNF stimulation, *Ripk1^{+/+}* BMDMs and MEFs showed increased level of tyrosine phosphorylation on RIPK1, but *Ripk1^{Y383F/Y383F}* mutation almost abolished tyrosine phosphorylation of RIPK1 (Fig. 4a and Supplementary Fig. 4a). Post translational modifications of RIPK1 plays a critical role for TNF-induced NF-κB activation, which could provide survival signal to suppress apoptosis and necroptosis activation[13,14,35]. However, we found

that nuclear translocation of p65 and phosphorylated IκBα, JNK and p38 in MEFs or BMDMs were not affected by *Ripk1^{Y383F/Y383F}* mutation in response to TNF (Fig. 4b, c and Supplementary Fig. 4b). Moreover, qPCR and ELISA analyses also showed inflammatory genes expression such as TNF, IL6, IL1β, and CXCL10 have no much difference between *Ripk1^{+/+}* and *Ripk1^{Y383F/Y383F}* BMDMs upon TNF stimulation (Fig. 4d and Supplementary Fig. 4c). These further suggest that *Ripk1^{Y383F/Y383F}* mutation has no effect on TNF-mediated inflammatory signaling activation.

Since JAK1 could phosphorylate RIPK1 on Y383 in TNF signaling, we next sought to determine whether tyrosine phosphorylation of RIPK1 involves in activation of JAK1 and STAT1. We found that the phosphorylation of JAK1 and STAT1 has no significant alteration in *Ripk1^{+/+}* and *Ripk1^{Y383F/Y383F}* BMDMs upon TNF, IFNγ or LPS stimulation (Supplementary Fig. 4d–f). These results suggest that JAK1 is activated upstream of RIPK1 and phosphorylates RIPK1 without affecting downstream activation of STAT1. In addition, pretreatment with IAP inhibitor BV6 could greatly promote TNF-mediated cell death in *Ripk1^{Y383F/Y383F}* BMDMs. However, we did not observe significant difference

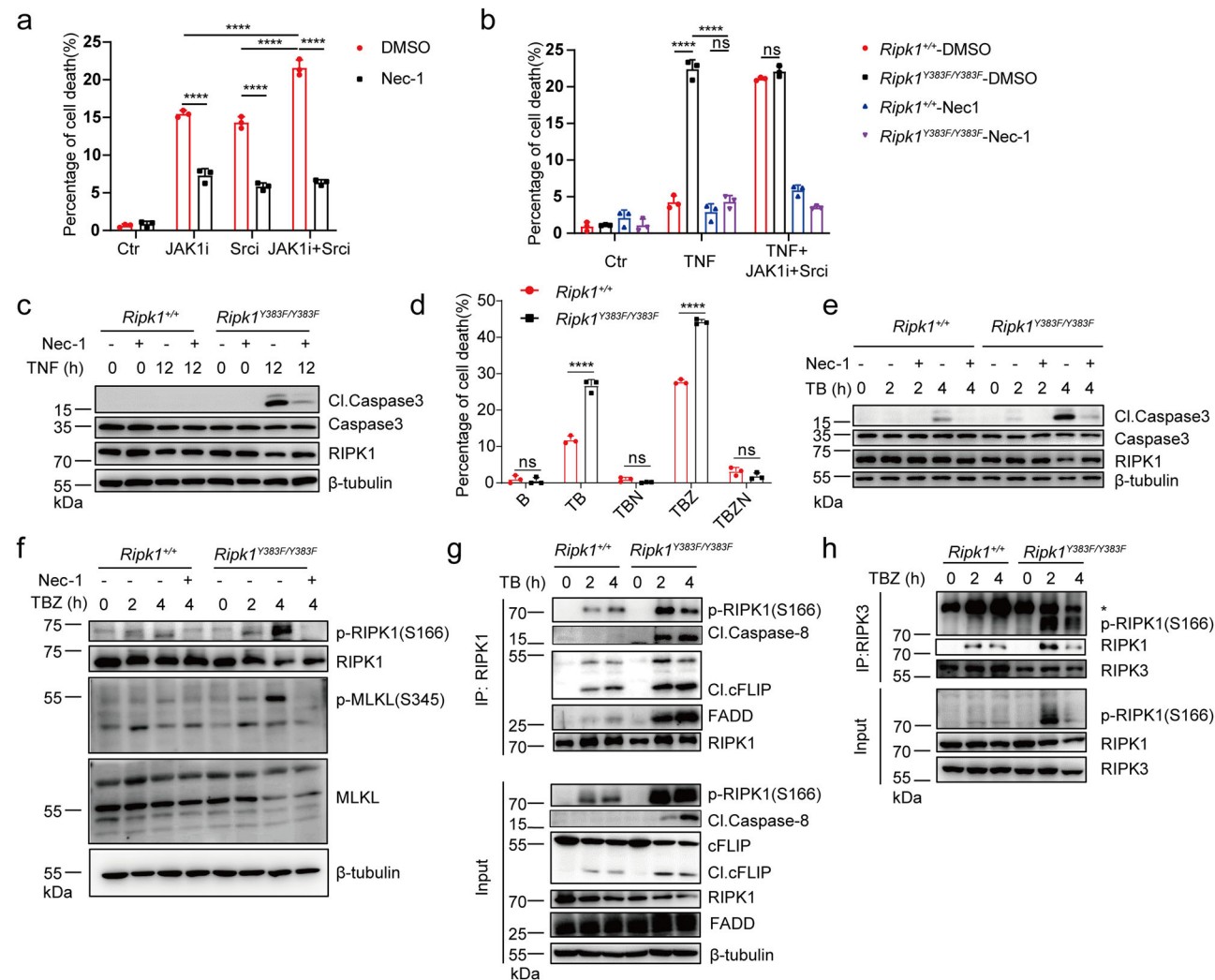

**Fig. 3 | Loss of tyrosine phosphorylation of RIPK1 promotes TNF-induced apoptosis and necroptosis. a** Primary WT BMDMs were stimulated by TNF with or without Nec-1 (RIPK1 inhibitor), JAK1 and SRC inhibitors for 24 h, and cell death were measured by SytoxGreen positivity. **b, c** Primary *Ripk1^{+/+}* and *Ripk1^{Y383F/Y383F}* BMDMs were stimulated by TNF with or without Nec-1 (RIPK1 inhibitor), JAK1 and SRC inhibitors for 24 h. Cell lysates were collected for western blotting (**c**) and cell death were measured by SytoxGreen positivity (**b**). **d** Primary *Ripk1^{+/+}* and *Ripk1^{Y383F/Y383F}* BMDMs were treated by different stimulators for 6 h. Cell death were measured by SytoxGreen positivity. T: TNF; B: BV-6; Z: zVAD.fmk; N: Necrostatin-1. Primary *Ripk1^{+/+}* and *Ripk1^{Y383F/Y383F}* BMDMs were stimulated by TNF/BV-6 (**e**) and

TNF/BV-6/zVAD (**f**) for indicated time points and whole-cell lysates were collected for western blotting. *Ripk1^{+/+}* and *Ripk1^{Y383F/Y383F}* immortalized MEFs were stimulated with TNF/BV-6 (**g**) or TNF/BV-6/zVAD (**h**) for indicated time points and whole-cell lysates were immunoprecipitated using anti-RIPK1 (**g**) or anti-RIPK3 (**h**) antibody. TNF: 100 ng/ml (**a**–**c**) and 10 ng/ml (**d**–**h**); BV-6: 2.5 uM; zVAD.fmk: 20 μM; Necrostatin-1: 10 μM; JAK1 inhibitor: 10 μM; SRC inhibitor: 10 μM. In **a**, **b**, **d**, data are represented as mean ± SEM (*n* = 3 independent cell samples for each genotype). Statistical significance was determined using a two-tailed unpaired t test. n.s. *p* > 0.05; ****p* < 0.0001. Source data are provided with this paper.

of TNF-induced proinflammatory genes expression in *Ripk1^{+/+}* and *Ripk1^{Y383F/Y383F}* BMDMs with BV6 treatment (Supplementary Fig. 4g). Thus, the enhanced cell death in *Ripk1^{Y383F/Y383F}* MEFs was not due to abnormality of TNF-induced pro-survival inflammatory signaling activation.

Moreover, we found that the ubiquitination and recruitment of RIPK1, A20, and TBK1 in TNF-RSC were unaffected in *Ripk1^{Y383F/Y383F}* MEFs upon TNF stimulation (Fig. 4e). However, we observed *Ripk1^{Y383F/Y383F}* mutation greatly impaired recruitment and phosphorylation of MK2 in TNF-RSC upon TNF stimulation (Fig. 4e). Since our previous results show JAK1 and SRC could regulate RIPK1 activity in TNF-RSC and complex II respectively, we also examined complex II formation and found that *Ripk1^{Y383F/Y383F}* mutation also impaired interaction between RIPK1 and MK2 in complex II (Fig. 4f). Consistently, the recruitment and phosphorylation of MK2 in TNF-RSC or complex II was also impaired in the deficiency of JAK1 or SRC (Supplementary Fig. 4h, i). These results

suggested that tyrosine phosphorylation of RIPK1 on Y383 is essential for MK2 binding and activation upon TNF stimulation.

MK2 has recently been demonstrated to suppress RIPK1 kinase-mediated apoptosis and necroptosis by phosphorylate RIPK1[22,24,26]. We, therefore, hypothesized that tyrosine phosphorylation of RIPK1 on Y383 inhibits RIPK1-dependent cell death in MK2-dependent manner. To test this hypothesis, we knocked down MK2 in both *Ripk1^{+/+}* and *Ripk1^{Y383F/Y383F}* immortalized MEFs and found deficiency of MK2 could largely diminish the difference of TNF-induced Caspase3 activation and cell death between *Ripk1^{+/+}* and *Ripk1^{Y383F/Y383F}* MEFs (Fig. 4g, h). Moreover, over-expression of constitutively active MK2 (MK2-CA) could largely suppress RIPK1 kinase activity and cell death in *Ripk1^{Y383F/Y383F}* MEFs (Fig. 4i, j). These results suggest that impaired MK2 recruitment and activation is a dominant but not the only cause of RIPK1 kinase activation in *Ripk1^{Y383F/Y383F}* cells. Since previous studies have

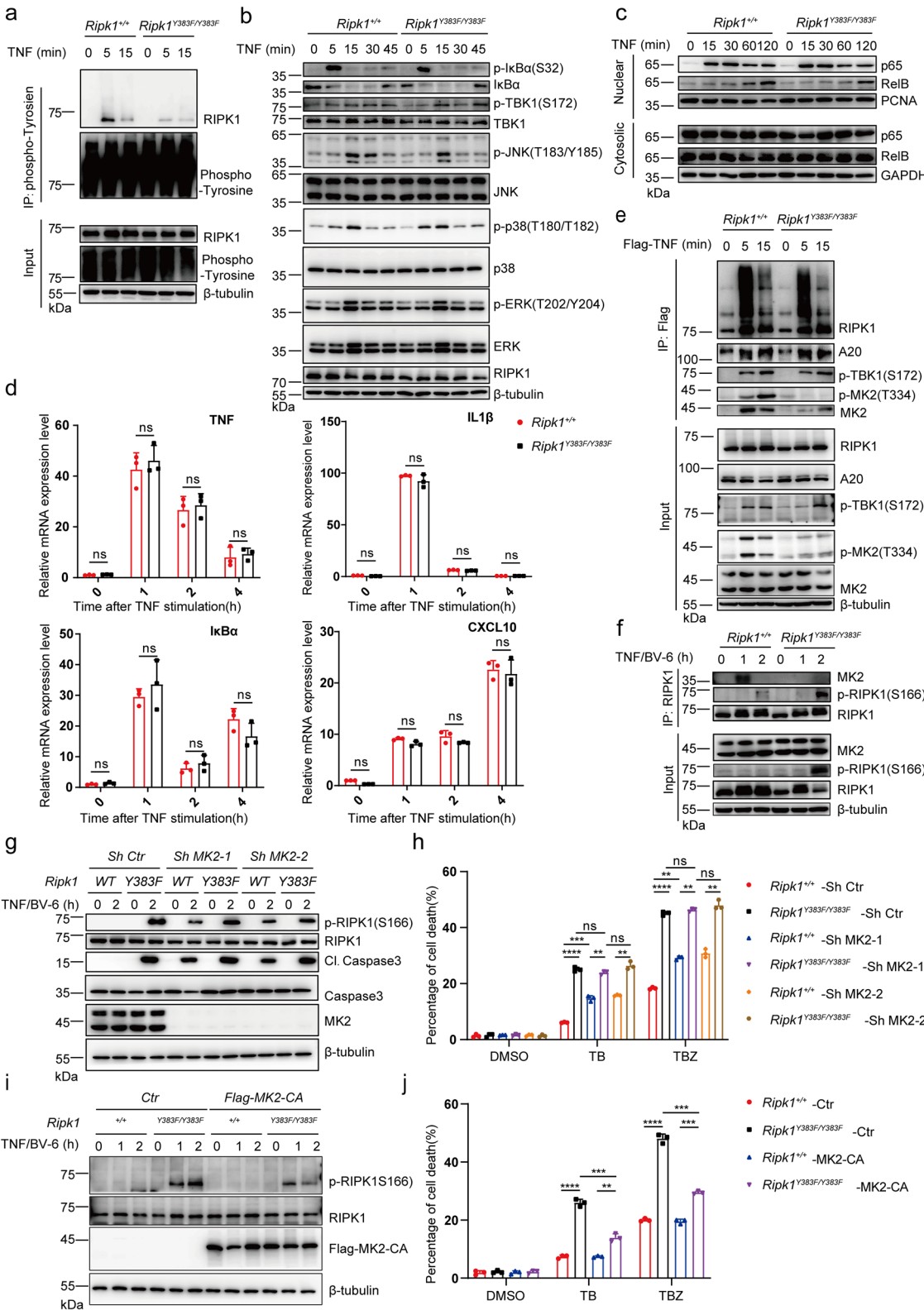

demonstrated that RIPK1 phosphorylation mediated by IKK and TBK1 could direct suppress RIPK1 kinase activity[20,23,25,36], we thus assume that tyrosine phosphorylation of RIPK1 might also has a directly inhibitory role on RIPK1 kinase activity besides activating MK2. Taken together, our results uncover that tyrosine phosphorylation of RIPK1 on Y383 enables MK2 activation and limit RIPK1 kinase activity to prevent TNF-induced apoptosis and necroptosis.

## RIPK1 Y383F mice develop systemic inflammation and emergency hematopoiesis

Phosphorylation of RIPK1 has been recently reported to play an essential role in TNF-induced cell death signaling, which could regulate inflammation, embryogenesis, and host defense against infections[19,22,25]. We next sought to characterize the physiological role of tyrosine phosphorylation of RIPK1 in *Ripk1*[Y383F/Y383F] mice. Although *Ripk1*[Y383F/Y383F] mice were viable and born at the expected Mendelian

**Fig. 4 | Tyrosine phosphorylation of RIPK1 is essential for limiting RIPK1 kinase activity. a** Primary *Ripk1*[+/+] and *Ripk1*[Y383F/Y383F] BMDMs were stimulated with TNF for indicated time points. Whole-cell lysates were immunoprecipitated with anti-phospho-tyrosine antibody for western blotting with indicated antibodies. **b**, **c** *Ripk1*[+/+] and *Ripk1*[Y383F/Y383F] immortalized MEFs were stimulated by TNF (10 ng/ml) for indicated time points. Cytosolic (**b**) and nuclear (**c**) extractions were collected for western blotting with indicated antibodies. **d** Primary *Ripk1*[+/+] and *Ripk1*[Y383F/Y383F] BMDMs were stimulated with TNF at different time point. The transcriptional and expression level of inflammatory NF-κB target genes were measured by qPCR. *Ripk1*[+/+] and *Ripk1*[Y383F/Y383F] immortalized MEFs were stimulated by Flag-TNF (100 ng/ml) (**e**) or TNF (10 ng/ml) /BV-6 (2.5 μM) (**f**) for indicated time points. Whole cell lysates were immunoprecipitated by anti-Flag resin (**e**) or anti-RIPK1 antibodies (**f**) for western blotting with indicated antibodies. *Ripk1*[+/+] and *Ripk1*[Y383F/Y383F] immortalized MEFs with or without knockdown of MK2 (**g**, **h**) or overexpression of constitutively active MK2 (MK2-CA) (**i**, **j**) were treated with TNF (10 ng/ml) /BV-6 (2.5 μM) for indicated time points. Whole-cell lysates were collected for western blotting (**g**, **i**). Cell death was measured by SytoxGreen positivity (**h**, **j**). T: TNF (10 ng/ml); B: BV-6 (2.5 μM); Z: zVAD.fmk (20 μM). In **d**, **h**, **j**, data are represented as mean ± SEM (*n* = 3 independent cell samples). Statistical significance was determined using a two-tailed unpaired t test. n.s. *p* > 0.05; ** *p* < 0.01; ***\*p* < 0.001; \*\*\*\**p* < 0.0001. Source data are provided with this paper.

ratios (Supplementary Fig. 5a), a decreased body weight was observed with the growth of *Ripk1*[Y383F/Y383F] mice (Fig. 5a). Notably, *Ripk1*[Y383F/Y383F] mice developed spontaneous inflammation and showed obvious splenomegaly by 8 weeks of age, coincident with an increased number of neutrophils (CD11b[+]Ly6G[+]) and monocytes (CD11b[+]Ly6C[+]) in the spleen (Fig. 5a, b). The proinflammatory cytokines IL6 and IL1β were elevated in the serum of *Ripk1*[Y383F/Y383F] mice (Supplementary Fig. 5b). Moreover, H&E staining and immuno-histochemical analysis showed that *Ripk1*[Y383F/Y383F] mice had more infiltrated CD11b, Ly6G, CD45 and F4/80 positive inflammatory cells in the liver (Fig. 5c and Supplementary Fig. 5c). Thus, these results indicate that deficiency of tyrosine phosphorylation of RIPK1 on Y383 induces systemic inflammation.

To further investigate the underlying cause of systemic inflammation and splenomegaly in *Ripk1*[Y383F/Y383F] mice, we analyzed hematopoiesis in bone marrow of *Ripk1*[Y383F/Y383F] mice. Notably, we observed elevated population of hematopoietic progenitor cells (LK, Lin[−]Sca-1[−]Kit[+]), hematopoietic stem and early progenitor cells (HSPCs, Lin[−]Sca-1[−]Kit[+] (LSKs)) and hematopoietic stem cells (HSCs, LSK-CD150[+]CD34[−]) in the bone marrow of *Ripk1*[Y383F/Y383F] mice (Fig. 5d). In addition, we found huge expansion of the granulocytes and monocytes progenitors (GMPs, Lin[−]Sca-1[−]Kit[+]FcγR[+]CD34[+]) in *Ripk1*[Y383F/Y383F] mice (Fig. 5e), which is consistent with enhanced mature neutrophils, inflammatory monocytes and macrophages in the bone marrow (Supplementary Fig. 5d). Conversely, the population of common myeloid progenitors (CMPs, Lin[−]Sca-1[−]Kit[+]FcγR[−]CD34[+]) and megakaryocyte-erythrocyte progenitors (MEPs, Lin[−]Sca-1[−]Kit[+]FcγR[−]CD34[−]) were greatly reduced in *Ripk1*[Y383F/Y383F] mice compared to littermate controls (Fig. 5e). Likewise, reduced numbers of erythroid lineage Ter119[+] cells and different erythroblast populations were observed in *Ripk1*[Y383F/Y383F] bone marrows (Supplementary Fig. 5e, f). Interestingly, Ter119[+] erythrocytes were markedly expanded in spleen, suggesting increased numbers of erythrocytes might contribute to the splenomegaly in *Ripk1*[Y383F/Y383F] mice (Supplementary Fig. 5g). These indicate that defect of tyrosine phosphorylation of RIPK1 on Y383 causes sustained emergency hematopoiesis as shown by dysregulated HSCs proliferation and differentiation towards myeloid lineage.

Recent studies have shown an important interplay between inflammation and hematopoiesis[37]. To further determine whether the systemic inflammation in *Ripk1*[Y383F/Y383F] mice was hematopoietic cell intrinsic, we generated and examined bone marrow chimeras of *Ripk1*[+/+] and *Ripk1*[Y383F/Y383F] mice until the donor-derived hematopoiesis was well established. As expected, no signs of inflammation were observed in the mice reconstituted with *Ripk1*[+/+] bone marrows, whereas mice reconstituted with *Ripk1*[Y383F/Y383F] bone marrows showed splenomegaly alone with massive infiltration of neutrophils in the spleen (Fig. 5f, g). Therefore, these results imply that *Ripk1*[Y383F/Y383F] mutation in the hematopoietic compartment is indispensable and sufficient to trigger the systemic inflammation in *Ripk1*[Y383F/Y383F] mice.

## TNF-induced cell death is responsible for systemic inflammation and emergency hematopoiesis in RIPK1 Y383F mice

TNF-TNFR1 signaling has been widely known as a master regulator in inflammation and hematopoiesis[38]. We then hypothesize that TNF-TNFR1 signaling is essential for the systemic inflammation in *Ripk1*[Y383F/Y383F] mice. We crossed *Tnfr1*[−/−] mice with *Ripk1*[Y383F/Y383F] mice and found that the inflammatory neutrophil infiltration in the spleen of *Ripk1*[Y383F/Y383F] *Tnfr1*[−/−] mice were comparable to *Tnfr1*[−/−] littermate controls (Fig. 6a). Furthermore, immunohistochemical staining results showed no more infiltration of inflammatory immune cells in the liver section of *Ripk1*[Y383F/Y383F] *Tnfr1*[−/−] mice (Fig. 6b). In addition, we also found that the population of hematopoietic progenitors LKs, LSKs, HSCs, GMPs, and MEPs had no significant difference between *Ripk1*[Y383F/Y383F]*Tnfr1*[−/−] mice and *Tnfr1*[−/−] littermate controls (Fig. 6c, d). Thus, our results indicate that TNF/TNFR1-induced downstream signaling triggers systemic inflammation in *Ripk1*[Y383F/Y383F] mice.

Loss of tyrosine phosphorylation of RIPK1 on Y383 promotes RIPK1 kinase-dependent apoptosis and necroptosis, which has been widestudied in the pathogenesis of inflammatory diseases[39,40]. To further explore the role of RIPK1 kinase activity in the systemic inflammation in *Ripk1*[Y383F/Y383F] mice, we treated *Ripk1*[+/+] and *Ripk1*[Y383F/Y383F] mice with RIPK1 kinase inhibitor Nec-1s and found RIPK1 inhibition could ameliorate the splenomegaly in *Ripk1*[Y383F/Y383F] mice (Supplementary Fig. 6a). Consistently, Nec-1s treatment could largely rescue infiltration of neutrophils in the spleen of *Ripk1*[Y383F/Y383F] mice (Fig. 6e). Further analysis of the hematopoiesis in the bone marrow revealed that the disorders of hematopoietic progenitor GMPs and MEPs in *Ripk1*[Y383F/Y383F]mice were largely rescued by RIPK1 kinase inhibition (Supplementary Fig. 6b). These results suggest the critical role of RIPK1 kinase activation in the pathogenesis of systemic inflammation in *Ripk1*[Y383F/Y383F]mice.

To determine whether aberrant cell death caused systemic inflammation in *Ripk1*[Y383F/Y383F] mice, we blocked downstream cell death signaling in *Ripk1*[Y383F/Y383F] mice by genetically co-deleting RIPK3 and Caspase8. Since RIPK3 and Caspase8 double deficient mice suffer from lymphadenopathy and splenomegaly at an old age[41,42], we compared *Ripk1*[+/+], *Ripk1*[Y383F/Y383F], *Ripk3*[−/−]*Caspase8*[−/−] and *Ripk1*[Y383F/Y383F] *Ripk3*[−/−]*Caspase8*[−/−] mice side by side at a young age at 8 weeks and old age at 16 weeks. At age of 8 weeks, the enlarged spleen in *Ripk1*[Y383F/Y383F] mice could be fully rescued by co-deletion of RIPK3 and Caspase8 (Supplementary Fig. 6c). Accordingly, deficiency of RIPK3 and Caspase-8 completely rescued infiltration of inflammatory immune cells in the spleen and liver of *Ripk1*[Y383F/Y383F] mice at age of 8 weeks (Fig. 6f, g). Further analysis of the hematopoiesis in the bone marrow revealed that the hematopoietic progenitor LKs, LSKs, HSCs, GMPs, and MEPs in *Ripk1*[Y383F/Y383F]*Ripk3*[−/−]*Caspase-8*[−/−] mice were comparable to the *Ripk3*[−/−]*Caspase-8*[−/−] littermate controls (Fig. 6h, i). At age of 16 weeks, similar as RIPK3 and Caspase8 deficient mice, the *Ripk1*[Y383F/Y383F]*Ripk3*[−/−]*Caspase-8*[−/−] mice also showed dramatic splenomegaly which is caused by massive infiltration of CD3[+]B220[+] lymphocytes (Supplementary Fig. 6d, e). Nonetheless, there were no more enhanced infiltration of inflammatory immune cells in the spleen of *Ripk1*[Y383F/Y383F] *Ripk3*[−/−]*Caspase-8*[−/−] mice compared to *Ripk1*[Y383F/Y383F] mice (Supplementary Fig. 6f). Therefore, our results demonstrate that the systemic inflammation in *Ripk1*[Y383F/Y383F] mice is triggered by TNF-induced cell death.

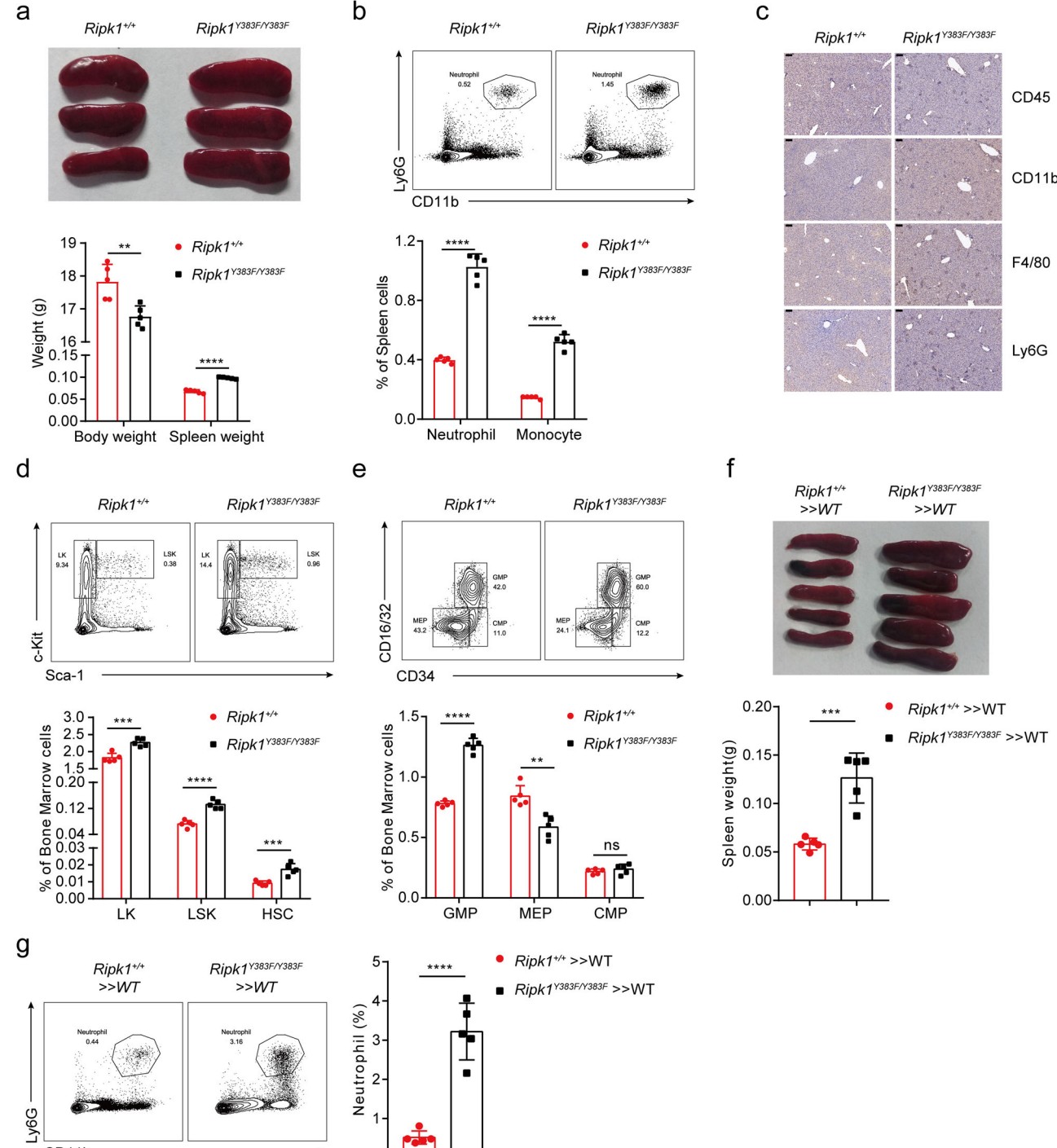

**Fig. 5 | RIPK1 Y383F mice develop systemic inflammation and emergency hematopoiesis. a** Representative images of spleen tissue and statistical results of spleen weight from *Ripk1*[+/+] (*n* = 5) and *Ripk1*[Y383F/Y383F] (*n* = 5) female mice at age of 6 weeks. **b** Flow cytometry and statistical analysis of neutrophils (CD11b[+]Ly6G[+]) and monocytes (CD11b[+]Ly6C[+]) in the spleen of *Ripk1*[+/+] (*n* = 5) and *Ripk1*[Y383F/Y383F] (*n* = 5) mice at 8 weeks of age. **c** Immunohistochemical staining of CD11b, Ly6G, CD45 or F4/80 in liver sections of *Ripk1*[+/+] and *Ripk1*[Y383F/Y383F] littermate mice at age of 8 weeks (Scale bar, 100 μm). Flow cytometry and statistical analysis in bone marrow for the indicated hematopoietic populations: LKs, LSKs, HSCs (**d**) and GMPs, MEPs, CMPs (**e**) from *Ripk1*[+/+] (*n* = 5) and *Ripk1*[Y383F/Y383F] (*n* = 5) mice at age of 8 weeks. **f** Images of spleen tissue and statistical analysis of spleen weight in *Ripk1*[+/+] » WT (*n* = 5) and *Ripk1*[Y383F/Y383F] » WT (*n* = 5) transplant mice at age of 8 weeks. **g** Flow cytometry and statistical analysis of neutrophils in the spleen from *Ripk1*[+/+] » WT (*n* = 5) and *Ripk1*[Y383F/Y383F] » WT (*n* = 5) transplant mice at age of 8 weeks. Data are represented as mean ± SEM. Statistical significance was determined using a two-tailed unpaired t test. n.s. *p* > 0.05; **\*\*\****p* < 0.01; **\*\*\****p* < 0.001; **\*\*\*\****p* < 0.0001. Source data are provided with this paper.

## Discussion

The importance of serine/threonine phosphorylation of RIPK1 mediated by TAK1, p38/MK2, TBK1/IKKε, and IKKα/β in suppressing RIPK1 kinase-mediated cell death is well established[19–26]. In this study, we found that tyrosine kinases JAK1 and SRC could mediate tyrosine phosphorylation of RIPK1 on Y383. Loss of tyrosine phosphorylation of RIPK1 on Y383 promotes RIPK1 kinase activation, which is partially due to impaired recruitment and activation of MK2. Hyperactivation of

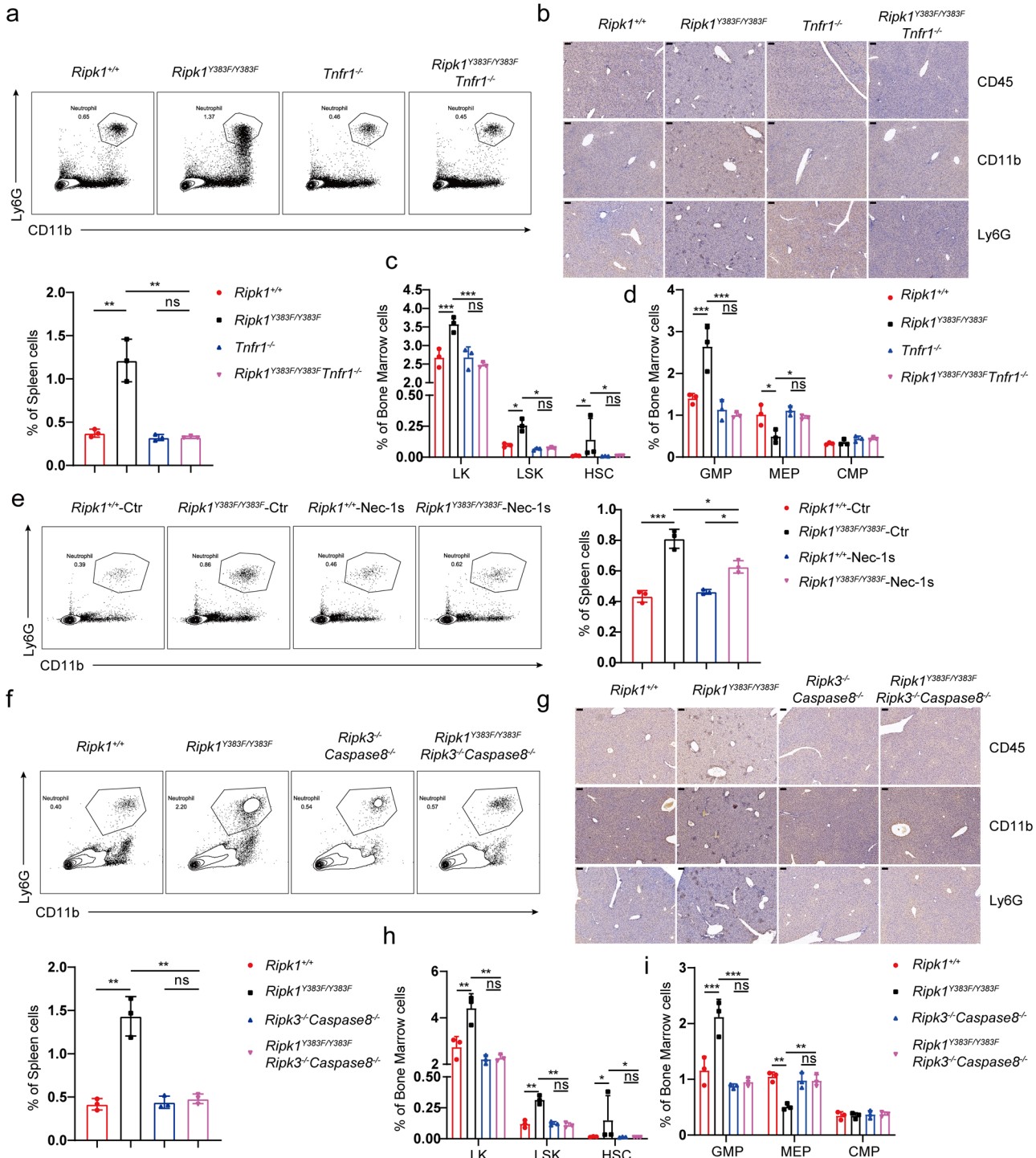

**Fig. 6 | TNF-induced cell death is responsible for systemic inflammation and emergency hematopoiesis in RIPK1 Y383F mice. a** Flow cytometry and statistical analysis for neutrophils (CD11b⁺Ly6G⁺) in the spleen from *Ripk1*⁺/⁺, *Ripk1*^Y383F/Y383F, *Tnfr1*⁻/⁻ and *Ripk1*^Y383F/Y383F *Tnfr1*⁻/⁻ mice at age of 8 weeks (*n* = 3). **b** Immunohistochemical staining of CD11b, Ly6G, and CD45 of liver sections from *Ripk1*⁺/⁺, *Ripk1*^Y383F/Y383F, *Tnfr1*⁻/⁻ and *Ripk1*^Y383F/Y383F *Tnfr1*⁻/⁻ mice at age of 8 weeks (Scale bar, 100 µm). Flow cytometry and statistical analysis in bone marrow for the indicated hematopoietic populations: LKs, LSKs, HSCs (**c**) and GMPs, MEPs, CMPs (**d**) from *Ripk1*⁺/⁺, *Ripk1*^Y383F/Y383F, *Tnfr1*⁻/⁻ and *Ripk1*^Y383F/Y383F *Tnfr1*⁻/⁻ mice at age of 8 weeks (*n* = 3). **e** *Ripk1*⁺/⁺ and *Ripk1*^Y383F/Y383F mice were treated with Nec-1s (3 mg/kg) every one day after 3 weeks old (*n* = 3). After 8 weeks, neutrophils (CD11b⁺Ly6G⁺) in the spleen from the mice were analyzed

by flow cytometry. **f** Flow cytometry and statistical analysis for neutrophils (CD11b⁺Ly6G⁺) in the spleen from *Ripk1*⁺/⁺, *Ripk1*^Y383F/Y383F, *Ripk3*⁻/⁻ *Caspase8*⁻/⁻ and *Ripk1*^Y383F/Y383F *Ripk3*⁻/⁻ *Caspase8*⁻/⁻ mice at age of 8 weeks (*n* = 3). **g** Immunohistochemical staining of CD11b, Ly6G, and CD45 of liver sections from *Ripk1*⁺/⁺, *Ripk1*^Y383F/Y383F, *Ripk3*⁻/⁻ *Caspase8*⁻/⁻ and *Ripk1*^Y383F/Y383F *Ripk3*⁻/⁻ *Caspase8*⁻/⁻ mice at age of 8 weeks (Scale bar, 100 µm). **h, i** Flow cytometry and statistical analysis in bone marrow for the indicated hematopoietic populations: LKs, LSKs, HSCs (**h**) and GMPs, MEPs, CMPs (**i**) from *Ripk1*⁺/⁺, *Ripk1*^Y383F/Y383F, *Ripk3*⁻/⁻ *Caspase8*⁻/⁻ and *Ripk1*^Y383F/Y383F *Ripk3*⁻/⁻ *Caspase8*⁻/⁻ mice at age of 8 weeks (*n* = 3). Data are represented as mean ± SEM. Statistical significance was determined using a two-tailed unpaired t test. n.s. *p* > 0.05; *\**p* < 0.05; *\*\**p* < 0.01; *\*\*\**p* < 0.001. Source data are provided with this paper.

RIPK1-dependent apoptosis and necroptosis leads to systemic inflammation and emergency hematopoiesis in RIPK1 Y383F mutant mice (Supplementary Fig. 7). However, impaired activation of MK2 only partially accounts for the hyperactivation of RIPK1 kinase in *Ripk1*$^{Y383F/Y383F}$ cells. Thus, the additional mechanisms how tyrosine phosphorylation directly or indirectly suppresses RIPK1 kinase activity still need further investigations. In addition, our results suggest JAK1 could be recruited to TNF-RSC to modulate RIPK1 activity. However, how JAK1 is activated and recruited to RIPK1 is still unclear. Interestingly, JAK1 could phosphorylate RIPK1 on Y384 residue, which is very close to K377 residue, a critical site for RIPK1 K63 ubiquitination and kinase inhibition[13,14]. Thus, we assume that the motif around K377 might determine RIPK1 ubiquitination status and interaction with JAK1.

Previous studies have indicated that multiple kinases could simultaneously phosphorylate RIPK1 and might act on the same phosphorylation site to suppress kinase activity of RIPK1[19–26]. JAK1 belongs to Janus kinase family, which contains JAK1, JAK2, JAK3, and TYK2. JAK3 and Tyk2 are mainly expressed on hematopoietic cells, whereas the function of JAK1 and JAK2 are more complicated due to their ubiquitous expression[43]. Therefore, other JAK family member such as JAK2 might also has a similar function as JAK1 to act on RIPK1 to modulate TNFR1-mediated cell death. In addition, whether the transcriptional activity of TNF-induced JAK1-STAT1 participates in TNFR1-mediated apoptosis and necroptosis signaling needs further investigation. Of note, other tyrosine kinase inhibitors such BTK and BCR-ABL inhibitors have been clinically used to treat B-cell malignancies and chronic myeloid leukemia by inducing efficient cancer cell death[44,45]. Therefore, whether these non-receptor tyrosine kinases could also tyrosine phosphorylate RIPK1 on Y384 to modulate RIPK1 activity in TNF-induced cell death signaling still needs further investigation.

The synergistical effect of interferon and tumor necrosis factor signaling has been widely studied in immune functions[46,47]. In addition, IFN signaling-mediated up-regulation of PKR, MLKL and ZBP1 has also been reported to modulate TNF-induced cell death, especially necroptosis signaling[48–50]. More recently, a study showed that IFNγ and TNF synergistically activate JAK-STAT1-IRF1 axis to promote nitric oxide production and driving caspase-8/FADD–mediated inflammatory cell death, which triggers tissue damage and mortality during SARS-CoV-2 infection in human[51]. However, these regulations of interferon signaling on TNF-triggered cell death are mainly in a transcription-dependent manner. Moreover, JAK1 inhibition could block IFNγ-induced necroptosis activation but promote TNF-induced apoptosis and cell death in our results, seemingly suggesting a contradictory role of JAK1 in interferon and TNF signaling[50]. However, one possibility is that the IFN-JAK1 activation could compete and inhibit TNF-JAK1 signaling to promote TNF-induced cell death, which still needs further investigation. Together, these results provide a potential new insight on the synergism of interferon and TNF signaling through JAK1 in a transcription-independent manner.

The importance of MK2 in phosphorylation-mediated suppression of RIPK1 kinase activity to preventing TNF-induced cell death has been well established[22,24,26]. However, how MK2 associates with RIPK1 and promotes self-activation to phosphorylate RIPK1 has been poorly understood. Herein, our results suggest that tyrosine phosphorylation of RIPK1 on Y383 is important for MK2 binding with RIPK1 and further activation of MK2, which provides a new insight how MK2 associates with RIPK1 and regulates RIPK1 activity. However, the exact mechanism how tyrosine phosphorylation of RIPK1 leads to MK2 activation remains unknown so far. Phosphorylated tyrosine residue of substrates is well known for binding with SH2 domain-containing proteins to activate downstream signaling. However, structural prediction in database suggests that MK2 is unlike to have SH2 domain for phosphotyrosine residue binding. Thus, other SH2 domain-containing proteins might serve as an adaptor to bridge MK2 binding with tyrosine phosphorylated RIPK1. It is noteworthy JAK1 and SRC activation could activate SH2 domain-containing proteins such as STAT1 and STAT3, which have also been reported to involved in TNFR1 signaling[30,31]. Thus, the potential role of STAT1 and STAT3 in bridging RIPK1 phosphorylation and MK2 activation will be of great interest and needs further investigation.

Importantly, loss or gain of function of JAK1 and SRC caused by rare mutations has been reported to trigger combined immunodeficiency, cancer progression, or autoimmune disease in human patients[52–55]. In addition, polymorphisms in human RIPK1 are also recently reported to cause a wide spectrum of autoinflammatory diseases[15,16,56]. Thus, uncovering the precise physiological biology and molecular mechanism of tyrosine phosphorylation of RIPK1 will be of great benefit for designing new therapeutic approaches in related human diseases. By using biochemical and genetic methods, we uncover that tyrosine phosphorylation of RIPK1 on Y383 in mice could serve as a physiological brake to limit TNF-induced RIPK1-dependent cell death, thereby restraining systemic inflammation. Taken together, our investigations provide a potential target for pharmacological intervention, and mice bearing *Ripk1*$^{Y383F/Y383F}$ mutation could be a useful tool to explore the function of RIPK1 tyrosine phosphorylation in various mouse models of related human disease.

## Methods

### Mice

All experimental and control mice were bred separately and housed in the specific pathogen-free (SPF) animal facilities in Tsinghua University (light/dark cycle 10 h:14 h, temperature 22–26 °C, humidity 40–70%). *Tnfr1*$^{-/-}$, *Ripk3*$^{-/-}$, and *Caspase-8*$^{-/-}$ mouse lines on a C57BL6/J background were provided by Dr. Bryant G Darnay (University of Texas MD Anderson Cancer Center). We generated *Ripk1*$^{Y383F/Y383F}$ knock-in mice on a C57BL/6 background by CRISPR-Cas9 technology. Briefly, we injected three mRNA into the embryos: 20 ng/µl Cas9 mRNA, 10 ng/µl sgRNA (single guide RNA), and 40 ng/µl oligo donor constructs. The sgRNA (see in Supplementary Table 2) that target the genome DNA region surrounding Y383 residue of RIPK1 was produced by in vitro transcription. We also designed donor construct (see in Supplementary Table 2) that containing Y to F mutation and several synonymous mutations within the donor to prevent the sgRNA- directed Cas9 from cleaving mutated RIPK1 genomic locus. The founder mice were genotyped by genomic DNA PCR and DNA sequencing (see in Supplementary Table 3). The mice were euthanized via carbon dioxide ($CO_2$) inhalation and confirmed dead before performing further experiments. All mouse experiments were performed in compliance with institutional guidelines and according to the protocol approved by the Institutional Animal Care and Use Committee of Tsinghua University.

### Antibodies and reagents

Recombinant mouse TNF was purchased from R&D (410-MT-010). CHX (C-6255) and Necrostatin-1 (N9037) were obtained from Sigma. zVAD.fmk (C1202) was obtained from Beyotime. Necrostatin 2 racemate (Nec-1s, S8641), JAK1 inhibitor filgotinib (S7605), SRC inhibitor Bosutinib (S1014), and BV-6 (S7597) were purchased from Selleckchem. Flag-mTNF (ALX-522-009-C050) was purchased from Enzo Life Science. The following antibodies were used in western blotting experiments: Antibodies against RIPK1 (3493, 1:1000), p-RIPK1 (31122, 1:1000), cleaved Caspase3 (9664, 1:1000), Caspase3 (9665, 1:1000), cleaved Caspase8 (9429, 1:1000), Caspase8 (4790, 1:1000), A20 (5630, 1:1000), p-IκBα (2859, 1:1000), p-p38 (9211, 1:1000), p38 (9228, 1:1000), p-JNK (9251, 1:1000), JNK (9252, 1:1000), p-TBK1 (5483, 1:1000), TBK1 (3504, 1:1000), JAK1 (50996, 1:1000), p-JAK1 (74129, 1:1000), STAT1 (14994, 1:1000), p-STAT1 (9167, 1:1000), SRC (2109, 1:1000), p-MK2 (3007, 1:1000), MK2 (3042, 1:1000), cFLIP (56343, 1:1000) and RelB (4954, 1:1000) were purchased from Cell Signaling Technology; Antibodies against RIPK3 (sc-374639, 1:500),

IκBα (sc-1643, 1:500), PCNA (sc-56, 1:500), FADD (sc-6036, 1:500) and p65 (sc-109, 1:500) were purchased from Santa Cruz Biotechnology; Antibodies against phospho-MLKL (ab196436, 1:1000) and MLKL (ab67942, 1:1000); Antibodies against β-tubulin (BE0025-100, 1:10,000), GFP (BE2002, 1:10,000), GAPDH (BE0023, 1:10,000), secondary horseradish peroxidase (HRP)-conjugated anti-rabbit (BE0108-100, 1:10,000) and anti-mouse antibodies (BE0107-100, 1:10,000) were obtained from Easy Bio; Antibody against phospho-tyrosine (4G10, 1:1000) were purchased from Millipore; Antibody against Flag (M20008, 1:10,000), HA (M20006, 1:10,000) were purchased from Abmart. The following antibodies are used for flow cytometry analysis: AF700 anti B220 (48-0452-80, 1:200), V450 anti-CD3 (48-0032-8246-0041-82, 1:200), Percp-cy5.5 anti-CD4 (46-0041-82, 1:200), FITC anti-CD8 (11-0081-82, 1:200), FITC anti-CD11b (11-0112-82, 1:200), PE anti-F4/80 (12-4801-82, 1:200), PE anti-c-Kit (12-1171-81, 1:200), Percp-cy5.5 anti-CD16/32 (45-0161-82, 1:200) and V500-conjugated Cell viability Dye(65-0866-18, 1:200) were purchased from eBioscience; Percp-cy5.5 anti-Ly6C (561,237, 1:200), V450 anti-Ly6G (560,603, 1:200), FITC anti-CD34 (553,733, 1:200) antibodies were from BD Bioscience; APC anti-CD71 (113,819, 1:200), PE anti-Ter119 (116,207, 1:200), PB anti-Sca-1 (108,120, 1:200), APC anti-CD150 (115,909, 1:200) and FITC anti-Lineage (133,313, 1:200) antibodies were from BioLegend.

## Generation of immortalized MEFs

*Ripk1*[+/+] and *Ripk1*[Y383F/Y383F] primary mouse embryonic fibroblasts (MEFs) were generated from E13.5 embryos. After removing the placenta, yolk sac, head, and the dark red organs, embryos were finely minced and digested for 20 min in 0.25% trypsin. Single cell suspension was then obtained by pipetting up and down the digested embryos. At passages 3 to 5, primary MEFs were immortalized by infection with SV40-T expressing lentivirus.

## Cell culture

HEK293T (purchased from ATCC) and MEF cells were cultured in DMEM medium (GIBCO) supplemented with 10% FBS, non-essential amino acids, sodium pyruvate, penicillin, streptomycin. Bone marrow derived macrophage cells (BMDMs) were prepared from tibia and femur of 6-weeks mice and cultured in DMEM medium supplemented with 20% M-CSF-containing conditional medium from L929 cells and 15% FBS for 7 days. All cells were cultured at 37 °C and 5% $CO_2$.

## Quantitative real-time PCR

Total RNA was isolated using TRIzol (Invitrogen) and reverse transcribed using SuperScript III (Invitrogen). Quantitative real-time PCR (qRT-PCR) was performed using Power SYBR Green PCR Master Mix (Genestar). Data were acquired by ABI 7500 Real-Time PCR system (Applied Biosystems). The amounts of transcript were normalized to those for glyceraldehyde 3-phosphate dehydrogenase. Melting curves were run to ensure amplification of a single product. The primers used are listed in Supplementary Table 4.

## Cell infection

A lentiviral supernatant was collected 48 h after co-transfection of expression-plasmids (Lenti-SV40-T-BSD, Lenti-CRISPR-MK2-Puro, and Lenti-MK2-CA-Flag-Puro) with packaging plasmids (psPAX2 and pMD2.G) into HEK293T cells. Viral supernatants were collected after 48 h, and target cells were incubated with the supernatant in the presence of polybrane for 8–12 h. After infection with virus, viral supernatants were replaced with fresh medium. After 24 h, infected cells were selected using blasticidin or puromycin. The infection efficiency was determined by using Western-blotting analysis. The primers of plasmids for generating MK2 deficient cell lines are listed in Supplementary Table 5.

## Analysis of cell death

MEFs were seeded the day before at 10,000 per well in duplicates in a 96-well plate. The next day, cells were pretreated with the indicated compounds for 60 min and then stimulated with the indicated concentration of mTNF in the presence of 5 μM SytoxGreen (Invitrogen, S34860). SytoxGreen intensity were measured at intervals of one hour using a Fluostar Omega fluorescence plate reader, with an excitation filter of 485 nm (SytoxGreen) and an emission filter of 520 nm (SytoxGreen). Percentage of cell death was calculated as (induced fluorescence-background fluorescence)/(maximum fluorescence-background fluorescence) × 100. The maximal fluorescence is obtained by full permeabilization of the cells by using TritonX-100. All cell death data were presented as mean±s.e.m. of 3 independent experiments.

## Western-blotting and immunoprecipitation

Cells were harvested and washed with cold PBS, and then were lysed with lysis buffer (50 mM HEPES, pH 7.4, 150 mM NaCl, 1% NP-40, 1 mM EDTA) plus protease inhibitor cocktail and 1 mM DTT, 1 mM NaF, 1 mM PMSF and 2 mM $Na_3VO_4$. Whole-cell lysates were collected by centrifuge at 18,000 × g for 15 min at 4 °C. Whole-cell lysates were then separated by SDS-PAGE and transferred onto PVDF membrane (Millipore). The membrane was incubated with primary antibodies and HRP-conjugated secondary antibodies. The protein expression was detected by using chemiluminescent substrate (Thermo Scientific).

For overexpression and endogenous immunoprecipitation, $6 \times 10^6$ MEF or 293T cells were seeded per condition in a 100 cm² petri dish. The next day, cells treated as indicated in figure legend were harvested and washed with cold PBS, and then lysed in lysis buffer (30 mM Tris-HCl, pH 7.4, 120 mM NaCl, 2 mM EDTA, 2 mM KCl, 10% glycerol, and 1% Triton X-100) containing 1 mM sodium orthovanadate, 1 mM sodium fluoride, 1 mM PMSF, and a protease inhibitor mixture (Roche). Whole-cell lysates were cleared by centrifugation (18,000 × g, 10 min, 4 °C). Immunoprecipitations were performed at 4 °C for 4 h with anti-Flag M2 beads (A2220; Sigma) for TNF-RSC IP, with RIPK1 antibodies (3493, Cell Signaling Technology) or RIPK3 antibodies (sc-374639, Santa Cruz) plus Protein G Sepharose 4FF (GE Healthcare) for Complex II IP. For over-expression immunoprecipitation, whole cell lysates of transfected HEK293T cells were collected and incubated with anti-Flag (M20008, Abmart) plus protein A/G agarose (A10001, Abmart) or with GFP-Nanoab-Agarose (GNA-20-400, Lablead) at 4 °C for 4 h. Beads were washed 4 times in lysis buffer and samples eluted by boiling in 10 μl 5× SDS loading dye. The Input and IP samples were then subjected to SDS-PAGE.

## Nuclear extraction

$3 \times 10^6$ MEF cells were lysed in nuclear lysis buffer (10 mM HEPES, pH 7.9, 10 mM KCl, 0.1 mM EDTA, and 0.4% NP40) containing protease inhibitors for 30 min on ice. The pellet (intact nuclei) was collected by centrifuging at 18,000 × g for 15 min at 4 °C and washed in nuclear lysis buffer. After washing for at least 5 times, the pellet was lysed in nuclear extraction buffer (20 mM HEPES, pH 7.9, 0.4 M NaCl, and 1 mM EDTA) containing protease inhibitors for 2 h at 4 °C. The nuclear extracts were collected in the supernatants by centrifuging at 18,000 × g for 15 min at 4 °C.

## Bone marrow chimeras

Bone marrow cells were collected by flushing the femur and tibias of CD45.2[+] mice with PBS using a 22.5-gauge needle. The cell suspension was treated with red blood cell lysis buffer and resuspended in PBS. Donor cells ($0.5 \times 10^6$ per genotype per mouse) were transplanted via intravenous injection into lethally irradiated (2 × 550 rad) sex-matched congenic CD45.1[+] recipient mice. Reconstitution was monitored 6–8 weeks after transplantation.

## Flow cytometry analyses

The single-cell suspension isolated from the spleen, bone marrow, and thymus of mice was used for staining cell surface markers following standard protocols. The gating strategy is provided in Supplementary Information (see in Supplementary Fig. 8). Antibodies for flow cytometry were used with 200-fold dilution. Data acquisition was performed using FACSAria II cytometer (BD). Flow cytometric data were analyzed with the FlowJo software.

## Histopathology, Immunohistochemistry

For histopathology analyses, liver was fixed in 4% paraformaldehyde solution, processed according to standard procedures, embedded in paraffin, and sectioned. Five micrometers-thick sections were stained with immunohistochemistry. We used the following antibodies: F4/80 (Ceville), Ly6G (Ceville), CD45 (Ceville), CD11b (Ceville). And images were acquired using light microscopy.

## Mass spectrometry analysis

$1 \times 10^7$ HEK293T were seeded in ten $150 \, cm^2$ petri dishes per condition. The next day, cells were over-expressed FLAG-hRIPK1 and then immunoprecipitated according to the TNFR1 complex1 IP protocol (see in immunoprecipitation section). After the final wash step in TritonX-100 lysis buffer, the beads were additionally washed five times in ice-cold PBS. And then the immunocomplexes were separated by SDS-PAGE gel. The gel bands of interest were excised from the gel, reduced with 25 mM of DTT and alkylated with 55 mM iodoacetamide which was followed by in-gel digestion with sequencing grade modified trypsin at 37 °C overnight. The peptides were extracted twice with 0.1% trifluoroacetic acid in 50% acetonitrile aqueous solution for 30 min and then dried in a speedvac. Peptides were redissolved in 25 µl 0.1% trifluoroacetic acid and 6 µl of extracted peptides were analyzed by Thermo orbitrap fusion.

For LC-MS/MS analysis, the peptides were separated by a 60 min gradient elution at a flow rate 0.30 µl/min with EASY-nLC 1000 system, which was directly interfaced with an Orbitrap Fusion Tribrid mass spectrometer (Thermo Fisher Scientific, Bremen, Germany). The analytical column was a home-made fused silica capillary column (75 µm ID, 150 mm length; Upchurch, Oak Harbor, WA) packed with C-18 resin (300 Å, 5 µm, Varian, Lexington, MA). Mobile phase consisted of 0.1% formic acid, and mobile phase B consisted of 100% acetonitrile and 0.1% formic acid. The Orbitrap Fusion mass spectrometer was operated in the data-dependent acquisition mode using Xcalibur3.0 software and there was a single full-scan mass spectrum in the orbitrap (350–1550 m/z, 120,000 resolution) followed by top-speed MS/MS scans in the Orbitrap.

The MS/MS spectra from each LC-MS/MS run were searched against the target protein database from UniProt proteome (human20191231) using an in-house Proteome Discoverer (Version PD1.4, Thermo-Fisher Scientific, USA). The search criteria were as follows: full chymotrypsin specificity was required; four missed cleavage was allowed; carbamidomethylation (C) were set as the fixed modifications; the oxidation (M) and 54.01063 Da (F) was set as the variable modification; precursor ion mass tolerances were set at 20 ppm for all MS acquired in an orbitrap mass analyzer; and the fragment ion mass tolerance was set at 0.02 Da for all MS2 spectra acquired. The scan sequence began with an MS1 spectrum (Orbitrap analysis, resolution 15,000, Isolation window was set as 2.0 m/z, AGC target set as 1.00e5, dynamic exclusion was set as 15.0 s, TopN was set as 40). The gas phase fragmentation was done by HCD (Resolution 60,000, AGC target set as 3.00e6, Scan range set as 200–1800 m/z, and Maximum injection time was set as 20 ms). The minimum length of peptides considered for proteome analysis is 6 amino acids. The peptide false discovery rate (FDR) was calculated using Fixed value PSM validator provided by PD. When the $q$ value was smaller than 1%, the peptide spectrum match (PSM) was considered to be correct. FDR was determined based on

PSMs when searched against the reverse, decoy database. Peptides only assigned to a given protein group were considered as unique. The false discovery rate (FDR) was also set to 0.01 for protein identifications[57]. The unique peptides which are considered as potential interactor of RIPK1 are summarized in the supplementary files. We thank Xiaolin Tian and Dr. Haiteng Deng in Center of Biomedical Analysis, Tsinghua University, for MS analysis.

## Statistics and reproducibility

All values in this article were given as mean± s.e.m, unless stated otherwise. All experiments were reproduced at least three independent times, and results shown in this article were representative. Statistical significance was calculated by two-tailed unpaired t test using GraphPad Prism software. Statistical significance was set based on $p$ values. n.s. (no significance), $p > 0.05$; $*p < 0.05$; $**p < 0.01$; $***p < 0.001$; $****p < 0.0001$.

## Reporting summary

Further information on research design is available in the Nature Research Reporting Summary linked to this article.

## Data availability

The authors declare that the data supporting the findings of this study are available within the paper and its Supplementary Information Files. The RIPK1 interacting proteomic data generated in this study have been deposited to the ProteomeXchange Consortium via the PRIDE database[58]. Data are available via ProteomeXchange with the accession number PXD036995. Other source data are provided as source data files. The source data of statistical analysis in the figures are provided as a Source Data file 1. The source data of uncropped scans of Western-blots in the figures are provided as a Source Data file 2. Mass-spectrometry analysis refer to Uniprot database is provided as a Source Data File 3. Source data are provided with this paper.

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

## Acknowledgements

We thank Dr. Bryant G Darnay (University of Texas MD Anderson Cancer Center, Houston, USA) for providing $Tnfr1^{-/-}$, $Ripk3^{-/-}$, and $Caspase8^{-/-}$ mice. This work was partially supported by the grant from National Key Research and Development Program of China (2019YFA0508502 to Xin Lin), National Natural Science Foundation of China (31930039, 81630058, 91942303, 31821003 to Xin Lin, and 81971469, 31670904 to Xueqiang Zhao), and annual funding from Tsinghua University-Peking University Jointed Center for Life Sciences.

## Author contributions

H.T., X.Z., and X.L. conceived and designed the study. H.T. performed most of the experiments. W.X. and J.Z. provided help for some experiments. H.T., W.X., and X.L analyzed and interpreted data. H.T. and X.L. wrote the manuscript, with all authors contributing to the writing and providing feedback.

## Competing interests

The authors declare no competing interests.

## Additional information

**Correspondence and requests** for materials should be addressed to Xin Lin.

