## [Peer Review File · Nature Communications]

Tyrosine phosphorylation regulates RIPK1 activity to limit cell death and inflammationREVIEWER COMMENTS

Reviewer #1 (Remarks to the Author):

Tu et al report on the role of RIPK1 tyrosine phosphorylation in regulating of TNF-induced cell death. They identified tyrosine 383 as a functionally important phosphorylation site on murine RIPK1 that restricts TNFR1- Casp8- and RIPK3-dependent cell death. Furthermore, the study suggests two novel players in the regulation of RIPK1 kinase activity, JAK1 and SRC, as upstream kinases that promote tyrosine phosphorylation on RIPK1. The authors link the mechanism of protection by p-T383 to recruitment of MK2, a kinase with an established role in phosphorylating RIPK1 and restricting RIPK1 kinase-activity dependent cell death.

The characterization of the effect of tyrosine phosphorylation is of significance to the field as it expands our understanding of the regulatory mechanisms on RIPK1. The reported findings are novel and Ripk1Y383F/Y383F mice may present an interesting model to study deregulated RIPK1 kinase-dependent signalling, specifically induced by elimination of a tyrosine phosphorylation-dependent checkpoint on RIPK1. However, the conclusions drawn in some aspects lack robustness especially with regards to the employment of immortalized MEFs instead of primary cells for in vitro studies as well as the mechanistic role of MK2 in the protection mediated by p-T383. Furthermore, in contrast to the Ripk1Y383F/Y383F mice that develop mild systemic inflammation, MK2-deficient mice do not develop a similar inflammatory phenotype, arguing that the phenotype of Ripk1Y383F/Y383F mice cannot be explained by impaired MK2 recruitment.

Specific comments

Figure 1:

b: categorization of NEMO as a kinase is misleading; a complete list of the proteins detected should be provided to give the reader an idea of how specifically the experiment can detect bona fide RIPK1-interacting partners; what does PSM stand for?

g: It is surprising that the FLAG antibody does not seem to detect tyrosine-phosphorylated RIPK1, otherwise one would expect a shift in the Flag band in the last lane in the IP samples. Do the authors have any explanation for this?

Figure 2:

a: RIPK1 should be written in capital letters here as the figure refers to protein domains, instead of italics (*Ripk1*)

b: How do the authors explain that the size of the Flag band runs at a different height for RIPK1-KD in the IP and Input ? Why are there multiple bands for RIPK1-ID both in the IP and in the Input ?

a,b: It should be discussed what might be the motifs that promote the interaction between JAK1 and RIPK1

Figure 3:

General comment: it is unfortunate that instead of analysing primary cells from Ripk1Y383F/Y383F mice, the authors decided to analyse immortalized MEFs throughout the study. Primary cells cultures from Ripk1Y383F/Y383F mice such as MEFs, skin or lung fibroblasts or BMDMs could easily be established and provide a cellular in vitro system with endogenous RIPK1 levels without the risk of generating cellular artefacts by viral infection during the immortalization process. In fact the authors also used BMDMs for the NF- κ B activation analysis in Figure S4 but did not analyse cell death responses in these cells. The wt immortalized MEF cell line used in this study furthermore seems to show an altered response to cell death stimuli as compared to primary cells (see panel comments below), further questioning the use of this system. All experiments assessing cell death responses biochemically and in cell death assays in cells from Ripk1Y383F mice need to be repeated using primary cells.

a,b: wt immortalized MEFs die in response to TNF only, which in the literature is not described to induce cell death in primary MEFs; it seems that both JAK and SRC in a non-redundant manner protect from TNF-induced cell death, it should be tested if their effect is synergistic by treating with both a JAK and SRC inhibitor at the same time.

a,b,c: the layout of these panels should be adjusted to make them more comparable for the reader (i.e. treatment for the same amount of time, adjustment of Y axis to the same values); Nec1s treatment to check for dependency on RIPK1 kinase activity should be included.
d: there seems to be cleavage of caspase 3 in response to TNF alone in wt cells, which does not correspond to reports in the literature, also wt immortalized MEFs show caspase-3 cleavage without treatment; questioning the use of this cell line.
e: dependency of TBZ-induced death on RIPK1 kinase activity should be tested, the authors should ideally use more specific RIPK1 inhibitor Nec1s to test this instead of Nec1.
f: which is the specific band for cleavage of Caspase-3? It should be discussed why in Nec-1 treated conditions cells still show caspase-3 cleavage.
i: the immunoblot for p-RIPK1(S166) in the RIPK3 IP is cropped too narrow, please show a larger part of the gel above and below to allow a better assessment.

Figure 4:

a: it is difficult to understand which is the specific band for RIPK1 in the phosphor-tyrosine IP. If available, RIPK1-deficient cells should be used as negative control. Moreover, the blots seem to be overexposed.
f,e: the authors suggest that MK2 is directly recruited to the TNF-RSC complex, while previous studies suggested that MK2 phosphorylates RIPK1 in the cytoplasm (see Jaco et al 2017, RIPK1 is not required to be recruited to TNFR1 to be modified by MK2). In the experiment shown there seems to be non-specific binding of MK2 in non-treated Flag-IP samples to M2 beads, which renders a conclusion of the data difficult. The authors should present more robust data on this or tone down their conclusions on MK2 recruitment to TNFR1.
g: if MK2 is the mechanistic link that mediates the protective effect of Y383F phosphorylation on the potential of RIPK1 to induce cell death, why do Ripk1Y383F/Y383F cells show increased levels of Caspase-3 cleavage than wt cells treated with MK2 inhibitor? In light of this observation it seems unlikely that the mechanism of protection through p-Y383 is exclusively mediated by MK2. Furthermore, previous reports show that MK2 rather provides a secondary checkpoint on RIPK1 kinase activity, abrogation of which does not induce cell death in response to TNF alone (Jaco et al 2017, Dondelinger et al 2017), but only in situations where cells are sensitized to cell death. The authors should test a more direct effect of Y383F mutation on RIPK1 kinase activity (for example by evaluating auto-phosphorylation at S166) under MK2-deficient conditions.

Figure 5:

d,e: abbreviations such as LK, LSK, HSC, GMP, MEP and CMP should be explained
The authors should provide information on a possible phenotype in other organs of Ripk1Y383F/Y383F mice such as the skin and lung that are affected in models of deregulated RIPK1 signalling such as Sharpincpdm/cpdm mice

Figure 6:

To show the dependency of systemic inflammation and emergency haematopoiesis in Ripk1Y383F/Y383F mice on RIPK1 kinase activity, the authors could consider inhibiting RIPK1 kinase activity in these mice, for example by treatment with a small molecule inhibitor such as Nec1s.

In general, the manuscript would benefit from an English language spelling and grammar check.

Reviewer #2 (Remarks to the Author):

The authors investigated the role of tyrosine phosphorylation of RIPK1 in cell death and inflammation, using MEF cells or mice with Y383 mutation. They provide evidence that non-receptor tyrosine kinases JAK1 and SRC bind with RIPK1 and phosphorylate RIPK1 at Y384 when overexpression on 293T cells. MEFs with RIPK1 Y383F blocks the MK2 phosphorylation, but promotes the RIPK1 activation and cell death including apoptosis and necroptosis under different treatment. Mice with Y383F mutation display systemic inflammation and emergency hematopoiesis which the authors suggest that can inhibited by deleting either TNFR1 or both RIPK3 and Caspase-8.

Regulation of RIPK1 by tyrosine phosphorylation is an area that has not been explored and thus has the potential to be interesting. However, the evidence provided is far from sufficient to demonstrate convincingly the presence of Y383(4) phosphorylation on RIPK1, the signaling mechanism that mediates this phosphorylation or its functional importance.

Main issue:

- 1) The authors started to investigate the role of JAK1 and SRC in the regulation of RIPK1 activity but somehow ended up on TNF which lacks logic as IFN is supposed to be involved in activating JAK and SRC, not TNF.
- 2) The authors found that JAK1/ SRC bind with RIPK1 and phosphorylate RIPK1 at Y384 when overexpression on 293T cells. However, no result is provided for the binding or phosphorylation of endogenous RIPK1 by JAK1/SRC.
- 3) The data on MEFs that almost focus on cell death cannot fully support the inflammation result in vivo.
- 4) Y383F mutant promotes RIPK1 activation in complex-II but not affect the recruitment of RIPK1 in complex-I and the NFKB activation. Whether knockout of JAK1/ SRC affect the complex-I or complex-II similarly with Y383F mutant, whether JAK1/ SRC be recruited into complex-I or complex-II required investigation.
- 5) Y383F mutant block the recruitment and phosphorylation of MK2, the loss function of p-MK2 is major effect by the Y383F mutant. It is reported that MK2 directly phosphorylates RIPK1 S321/336 in the cytosol but not in TNFR1 complex I (PMID: 28920952). As MK2 phosphorylates RIPK1 is directly, the mechanism how Y383F affect the MK2 activation still unclear.
- 6) Y383F mutation induced systemic inflammation and emergency hematopoiesis in vivo. It is important to investigate if RIPK1 is activated in Y383F mutation mice, and whether the inflammation induced by Y383F mutation in mice is dependent on RIPK1 activation or could be inhibited by RIPK1 inhibitor.
- (7) It is also important to examine the inflammation level on immune cells, like BMDM or PMBC cells by Y383/384 phosphorylation.
- (8) Certain key experimental data are missing. E.g Figure 6 is about comparing Ripk1-Y383F mutation in *Tnfr1*^{-/-} or *ripk3*^{-/-}*casp8*^{-/-}, but none of the comparing data included Ripk1-Y383F alone in Figure 6!

Reviewer #3 (Remarks to the Author):

The manuscript by Tu et al. proposes that phosphorylation of RIPK1 at Y383 by the kinases SRC or JAK1 is required to limit cell death and inflammation. The authors argue that this tyrosine phosphorylation enables MK2 binding to RIPK1 and subsequent MK2-mediated phosphorylation to further inhibit the activity of RIPK1. The authors also present a knock-in model which shows splenomegaly and emergency haematopoiesis, implying that this checkpoint on RIPK1 is relevant in vivo to prevent overt inflammatory phenotypes.

Despite these positive aspects, the data presented is often confusing and, to a major extent, relies on overexpression studies in HEK293 cells. Moreover, the conclusions drawn from the data obtained seem to be an oversimplification of the actual mechanisms at play. Therefore, the study would immensely benefit not only of a conceptual revisiting but also of additional endogenous characterization of crucial interactions. The in vivo phenotype of the new knock-in mutant seems clear, yet given the proposed mechanism, it is quite surprising that it is so different from those of MK2 knock-out and RIPK1 S321A knock-in mice. Moreover, the rescue with the RIPK3-Casp8 double knock-out raises further concerns that need to be clarified.

Major points:

Figure 1 and 2

1. While it is understandable that the early validation of the mass spec data and study of the domain involved in the binding is performed by overexpression studies, endogenous interactions must be shown for novel interactors. This is the expected standard in the field.
 - Does endogenous RIPK1 bind to endogenous SRC and/or JAK1 and under which conditions, i.e. following which stimulus can these interactions be detected?

- Is the binding of SRC and that of JAK1 to RIPK1 mutually exclusive or do these three proteins form a trimeric complex?

2. The fact that the so called "ID" domain containing the RHIM domain is the domain required for the binding of RIPK1 to JAK1 is concerning. These constructs could bind to the full length RIPK1 that is (although at very low levels) still expressed in HEK293t via the reciprocal RHIM domain. The authors need to repeat these domain-mapping studies in HEK293 cells in which endogenous RIPK1 has been knocked out and cells have subsequently been reconstituted with the individual tagged constructs.

3. Figure 1D is redundant. The fact that overexpressed proteins co-localize does not add anything to this manuscript. Also, this panel only shows three cells which is far too low a number to be of any significance.

Figure 3

4. The authors show that treatment with JAKi or SRCi enhances TNF sensitivity. Similar sensitivity is also detected in the RIPK1 Y383F knock-in cells treated with TNF. The amount of cell death detected is very low and goes from 10% to a little more than 15% when cells are co-treated with the inhibitors and in the case of the RIPK1 Y383F knock-in cells merely reaching a maximum 22% of cell death.

- How are these differences biologically significant? They may well be statistically significantly different but how can such low differences be sufficiently high to be biologically meaningful. The titration of higher concentrations of TNF and perhaps the additional co treatment with Smac mimetic drugs is highly recommended to further investigate this point.

- The authors need to try both inhibitors in combination and, most importantly, they also need to treat the Y383F cells with the JAKi and/or SRCi.

Figure 4

5. The result shown in Figure 4A contradicts the main message of this manuscript. While lane 2 is clearly darker, it is impossible to detect any difference for the RIPK1 band. Also, the blots are heavily overexposed and it is therefore impossible to judge the quality of the immunoprecipitation. This needs to be repeated and higher quality data need to be provided.

6. Figure 4E needs to be done under endogenous conditions. The authors have the knock-in cells and the conditions under which MK2 and RIPK1 are bound have been extensively published by three different groups. There is no reason why this data should be presented by utilizing an overexpression system.

7. Figure 4D, F, G show blots for pMK2. Why is total MK2 not shown (see point above)? Also, in all these three panels loss of pMK2 at the complex coincides with the overall loss of phosphorylation in the lysates. This is rather unexpected and would suggest that the Y383F mutation is somehow impacting the activation of MK2 which, canonically, is known to be independent of RIPK1 kinase activity. This needs to be further investigated because it points towards a direct effect of the mutant on MK2 activation and/or stability which would mean that the mechanism would be quite different from the one currently proposed by the authors in the manuscript.

Figure 4G-H:

8. Are the Sh-RNA utilised pools? How many were tested? More than one should be included to avoid potential off-target effects.

- In figure 4G it is clear that the ShMK2 alone has an effect on cleavage of Casp3 as compared to ShCtr. Interestingly, while the cleavage of Casp3 is enhanced in the Y383F cells where the levels of MK2 have been depleted these are considerably lower than the levels observed in control cells. These results raise numerous questions. If the effect of the Y383 phosphorylation is via MK2, why is it that Y383F shows more cleavage of Casp3 in the absence of MK2? Why are they lower than in control cells? The mutant should sensitize to TNF. This seems to contradict the results of Fig 3E and Fig 4H where the knockdown of MK2 shows no further sensitization to the knock-in mutation. Do the authors envisage more than one modality of cell death at play which does not involve only cleavage of Casp3? This needs to be revisited. As it stands, the mechanism is .

Figure 6

9. Figure 6 should also include the data for the knock-in mice. It is impossible to draw any conclusion otherwise. More importantly, how do the authors justify the same number of neutrophils in the TNFR1 knock-out mouse as in the RIPK3-Casp8 double knock-out mouse (0.77 Vs 0.71, Figure 6B, F). Moreover, and most importantly, the RIPK3-Casp8 knock-out mouse suffers from lymphadenopathy and splenomegaly (akin Fas-lpr or FasL-gld mice). This is clearly RIPK1-independent as RIPK3- Casp8-RIPK1 triple knock-out mice also present with this phenotype. How can the levels of hematopoietic cells be normal in these mice? The 5 genotypes (including the

knock-in) need to be compared side by side. This includes the cell sorting with same gating and the plotting and the percentage normalization on the same graphs. This current representation of the data does not account for the underlying phenotype of the double knock-outs and is therefore misleading. The conclusion that RIPK1 Y383F-induced inflammation and cell death is rescued by co-deletion of Casp8 and RIPK3 cannot be drawn from the presented data and must be properly analyzed.

We would like to thank reviewers for their constructive comments. Based on their comments and critiques, we have performed a series of experiments to address their questions and re-organized our manuscript. As results, we have repeated and added new panels to support our conclusion that tyrosine phosphorylation of RIPK1 is essential for inhibiting RIPK1 kinase activity and cell death, as shown in Fig. 1d, 1e, 1f, 1i, 3a, 3b, 3c, 3d, 3e, 3f, 4a, 4d, 4e, 4g, 4h, 4i, 4j, 4k, 6a, 6b, 6c, 6d, 6e, 6f, 6g, 6h, 6i and supplementary 3a, 4c, 4d, 5b, 6a, 6b, 6c, 6d, 6e, 6f. In addition, we also provide the following point-to-point response to address reviewers' critiques. Therefore, we believe that we have addressed all of questions and concerns raised by reviewers, and hope our manuscript is now suitable for publication in Nature Communications.

Reviewer #1 (Remarks to the Author):

Tu et al report on the role of RIPK1 tyrosine phosphorylation in regulating of TNF-induced cell death. They identified tyrosine 383 as a functionally important phosphorylation site on murine RIPK1 that restricts TNFR1- Casp8- and RIPK3-dependent cell death. Furthermore, the study suggests two novel players in the regulation of RIPK1 kinase activity, JAK1 and SRC, as upstream kinases that promote tyrosine phosphorylation on RIPK1. The authors link the mechanism of protection by p-T383 to recruitment of MK2, a kinase with an established role in phosphorylating RIPK1 and restricting RIPK1 kinase-activity dependent cell death.

The characterization of the effect of tyrosine phosphorylation is of significance to the field as it expands our understanding of the regulatory mechanisms on RIPK1. The reported findings are novel and *Ripk1*Y383F/Y383F mice may present an interesting model to study deregulated RIPK1 kinase-dependent signaling, specifically induced by elimination of a tyrosine phosphorylation-dependent checkpoint on RIPK1. However, the conclusions drawn in some aspects lack robustness especially with regards to the employment of immortalized MEFs instead of primary cells for in vitro studies as well as the mechanistic role of MK2 in the protection mediated by p-T383. Furthermore, in contrast to the *Ripk1*Y383F/Y383F mice that develop mild systemic inflammation, MK2-deficient mice do not develop a similar inflammatory phenotype, arguing that the phenotype of *Ripk1*Y383F/Y383F mice cannot be explained by impaired MK2 recruitment.

RESPONSE: We appreciate your positive comments on our study and pointing out some critical issues to improve our manuscript. As suggested, we repeated the in vitro studies with primary BMDMs and showed similar results as in immortalized MEFs (Fig. 3a-f). Moreover, for the mechanistic role of MK2 in protection mediated by p-Y383 of RIPK1, we repeated the data by using two parallel MK2-deficient cell lines and found that MK2 deficiency is not the only cause of RIPK1 kinase activation caused by Y383F mutation. In addition, expression of constitutive activated MK2 in RIPK1 Y383F cells could alleviate but not fully inhibit RIPK1 kinase activity and TNF-induced cell death (Fig. 4h-k). Since previous studies have demonstrated that RIPK1 phosphorylation mediated by IKK and TBK1 could direct suppress RIPK1 kinase activity, suggesting that tyrosine phosphorylation of RIPK1 might also has a directly inhibitory role on

RIPK1 kinase activity besides MK2. In addition, MK2 deficiency could decreased inflammatory cytokine production (PMID: 29770701), however, RIPK1 Y383F mutation has a normal response to TNF-induced inflammatory signaling (Fig. 4b-e). These results and observations might explain that MK2-deficient mice do not develop mild inflammatory phenotype as RIPK1 Y383F mice.

Specific comments

Figure 1:

b: categorization of NEMO as a kinase is misleading; a complete list of the proteins detected should be provided to give the reader an idea of how specifically the experiment can detect bona fide RIPK1-interacting partners; what does PSM stand for?

RESPONSE: Thank you for your comment. We felt sorry for misleading you on the categorization, we have corrected the description. PSM is short for peptide-spectrum match (PSM), a spectrum that matches to a peptide sequence, which stands for the abundance of interacting partners of RIPK1. In addition, we have provided a complete list of RIPK1-interacting partners in supplementary files.

g: It is surprising that the FLAG antibody does not seem to detect tyrosine-phosphorylated RIPK1, otherwise one would expect a shift in the Flag band in the last lane in the IP samples. Do the authors have any explanation for this?

RESPONSE: Thank you for your comment. We did observe this phenomenon that tyrosine phosphorylation of RIPK1 in our over-expression system did not cause a mobility shift on RIPK1. Actually, we could not figure out the fundamental reason. One possibility is that the structure of tyrosine phosphorylation could not induce the type of mobility shift as serine or threonine phosphorylation mediated by TAK1, IKK or TBK1, etc.

Figure 2:

a: RIPK1 should be written in capital letters here as the figure refers to protein domains, instead of italics (*Ripk1*)

RESPONSE: Thank you for this suggestion. As suggested, we have changed to capital letters in Fig. 2a.

b: How do the authors explain that the size of the Flag band runs at a different height for RIPK1-KD in the IP and Input? Why are there multiple bands for RIPK1-ID both in the IP and in the Input?

RESPONSE: We felt sorry for misleading you on the data presentation. RIPK1 KD (kinase domain truncation) is 55kD, and ID (Intermediated domain) is about 35kD, we have corrected the

labeling of molecular weight (Fig. 2b). The multiple bands of RIPK1-ID-Flag in the IP and Input could be non-specific bands, as we used HA-RIPK1-ID and did not find the multiple bands and also has a specific interaction with Flag-JAK1 (See attached data shown below).

a,b: It should be discussed what might be the motifs that promote the interaction between JAK1 and RIPK1

RESPONSE: Thank you for your comment. In this study, we found that tyrosine kinases JAK1 and SRC could tyrosine phosphorylate RIPK1 to suppress TNF α -induced RIPK1-dependent cell death. Our results suggest JAK1 could phosphorylate RIPK1 on Y384 residue, which is very close to K377 residue, a critical site for RIPK1 K63 ubiquitination and kinase inhibition. Thus, we assume that the motif around K377 might determine RIPK1 ubiquitination status and interaction with JAK1. We added this discussion in our manuscript (Page 17, Line 320-327). However, the detailed structural motif may need a further study.

Figure 3:

General comment: it is unfortunate that instead of analysing primary cells from Ripk1Y383F/Y383F mice, the authors decided to analyse immortalized MEFs throughout the study. Primary cells cultures from Ripk1Y383F/Y383F mice such as MEFs, skin or lung fibroblasts or BMDMs could easily be established and provide a cellular in vitro system with endogenous RIPK1 levels without the risk of generating cellular artefacts by viral infection during the immortalization process. In fact the authors also used BMDMs for the NF-kB activation analysis in Figure S4 but did not analyse cell death responses in these cells. The wt immortalized MEF cell line used in this study furthermore seems to show an altered response to cell death stimuli as compared to primary cells (see panel comments below), further questioning the use of this system. All experiments assessing cell death responses biochemically and in cell death assays in cells from Ripk1Y383F mice need to be repeated using primary cells.

RESPONSE: Thank you for your critique. As suggested, we used primary WT BMDMs and JAK1 and Src inhibitor to explore the role of JAK1 and Src in TNF-induced cell death. Consistent with our previous results in MEFs, JAK1 or Src inhibition could both enhanced the cell death in BMDMs in response to TNF (Fig. 3a). Also, we isolated BMDMs from WT and Y383F littermate mice and found the level of cell death including TNF-induced apoptosis and necroptosis is greatly higher in Y383F cells (Fig. 3b-f). These biochemical and cell death assays in primary BMDMs further demonstrated that tyrosine phosphorylation of RIPK1 is critical for preventing TNF-induced cell death.

a,b: wt immortalized MEFs die in response to TNF only, which in the literature is not described to induce cell death in primary MEFs; it seems that both JAK and SRC in a non-redundant manner protect from TNF-induced cell death, it should be tested if their effect is synergistic by treating with both a JAK and SRC inhibitor at the same time.

RESPONSE: Thank you for your suggestion. We found primary WT BMDMs indeed have no

significant response to TNF-induced cell death (Fig. 3a and b). Thus, the sensitivity to TNF in MEFs might have been changed after immortalization. However, the enhanced cell death by JAK1 or Src inhibition has also been observed in primary BMDMs, further supporting our previous results in immortalized MEFs (Fig. 3a). Moreover, as suggested, we found primary WT BMDMs treated with JAK and SRC inhibitor at the same time indeed has a enhance cell death compared to treatment of JAK or SRC inhibitor alone (Fig. 3a). These results suggest that JAK and SRC has a non-redundant function in protection from TNF-induced cell death.

a,b,c: the layout of these panels should be adjusted to make them more comparable for the reader (i.e. treatment for the same amount of time, adjustment of Y axis to the same values); Nec1s treatment to check for dependency on RIPK1 kinase activity should be included.

RESPONSE: Thank you for your thoughtful comment. As suggested, we have rearranged the layout of panels of cell death assays (Fig. 3a and b). Moreover, we found Nec1 treatment could greatly inhibit TNF-induced cell death triggered by JAK or SRC inhibition and Y383F mutation (Fig. 3a and b). These results indicate that loss of tyrosine phosphorylation of RIPK1 promotes TNF-induced cell death depends on RIPK1 kinase activity.

d: there seems to be cleavage of caspase 3 in response to TNF alone in wt cells, which does not correspond to reports in the literature, also wt immortalized MEFs show caspase-3 cleavage without treatment; questioning the use of this cell line.

RESPONSE: Thank you for your comment. The basal level of caspase-3 cleavage in both WT and Y383F MEF cells is relatively low, which might be due to the antibody specificity, blocking conditions or cell lines. To avoid these, we isolated primary BMDMs and found Y383F BMDMs showed more caspase-3 cleavage under TNF stimulation (Fig. 3c).

e: dependency of TBZ-induced death on RIPK1 kinase activity should be tested, the authors should ideally use more specific RIPK1 inhibitor Nec1s to test this instead of Nec1.

RESPONSE: Thank you for your thoughtful comment. We treated WT and Y383F primary BMDMs with TBZ and found enhanced RIPK1 S166-mediated auto-phosphorylation, suggesting higher RIPK1 kinase activity by Y383F mutation. In addition, pre-treatment of RIPK1 inhibitor Nec-1 could fully prevent the RIPK1 auto-phosphorylation and downstream MLKL activation (Fig. 3f). These suggest that RIPK1 Y383F mutation promotes TBZ-induced cell death in a RIPK1 kinase-dependent manner.

f: which is the specific band for cleavage of Caspase-3? It should be discussed why in Nec-1 treated conditions cells still show caspase-3 cleavage.

RESPONSE: Thank you for your comment. Cleaved Caspase-3 (Asp175) antibody detects endogenous levels of the large fragment (17/19 kDa) of activated caspase-3 resulting from cleavage adjacent to Asp175 (CST: #9664). RIPK1 kinase could almost fully block RIPK-dependent apoptosis, however, cIAPs inhibitor (BV-6) could also partially blocks NF-kB

activation or promotes mitochondria-mediated intrinsic apoptosis, which is independent of RIPK1 kinase activity. These might cause slight caspase-3 cleavage with RIPK1 inhibitor Nec-1s treatment.

i: the immunoblot for p-RIPK1(S166) in the RIPK3 IP is cropped too narrow, please show a larger part of the gel above and below to allow a better assessment.

RESPONSE: Thank you for your comment. We have provided a larger part of the gel (Fig. 3h)

Figure 4:

a: it is difficult to understand which is the specific band for RIPK1 in the phosphor-tyrosine IP. If available, RIPK1-deficient cells should be used as negative control. Moreover, the blots seem to be overexposed.

RESPONSE: Thank you for your comment. Since we had previously freeze RIPK1-deficient embryo, and lack RIPK1-deficient mice in our hand currently, instead, we repeated this experiment in primary BMDMs from WT and Y383F mice and found that Y383F mutation greatly impairs tyrosine phosphorylation of RIPK1 in response to TNF (Fig. 4a).

f,e: the authors suggest that MK2 is directly recruited to the TNF-RSC complex, while previous studies suggested that MK2 phosphorylates RIPK1 in the cytoplasm (see Jaco et al 2017, RIPK1 is not required to be recruited to TNFR1 to be modified by MK2). In the experiment shown there seems to be non-specific binding of MK2 in non-treated Flag-IP samples to M2 beads, which renders a conclusion of the data difficult. The authors should present more robust data on this or tone down their conclusions on MK2 recruitment to TNFR1.

RESPONSE: Thank you for your comment. Previous work indeed demonstrated MK2-mediated phosphorylation on RIPK1 is in cytosol, however, the recruitment and activation in TNFR1 complexes is not clearly answered. In this work, we did detect the recruitment and activation of MK2 in both TNFR1 complex1 and complex2 (Fig. 4f, g). Our previous published work also detected MK2 recruitment to TNFR1 complex1 in WT cells, which is significantly decreased in RIPK1 K63 ubiquitination defective K376R mutant cells (PMID:31519887). These results suggest that MK2 could potentially also function in TNFR1 complex to regulate cell death signaling.

g: if MK2 is the mechanistic link that mediates the protective effect of Y383F phosphorylation on the potential of RIPK1 to induce cell death, why do Ripk1Y383F/Y383F cells show increased levels of Caspase-3 cleavage than wt cells treated with MK2 inhibitor? In light of this observation it seems unlikely that the mechanism of protection through p-Y383 is exclusively mediated by MK2. Furthermore, previous reports show that MK2 rather provides a secondary checkpoint on RIPK1 kinase activity, abrogation of which does not induce cell death in response to TNF alone (Jaco et al 2017, Dondelinger et al 2017), but only in situations where cells are sensitized to cell death. The authors should test a more direct effect of Y383F mutation on RIPK1 kinase activity (for example by evaluating auto-phosphorylation at S166) under MK2-deficient conditions.

RESPONSE: Thank you for your thoughtful comment. We generated two MK2-knockdown cell lines with good Knockdown efficiency to repeat these experiments, and found that MK2 knockdown in Y383F cell still has higher caspase-3 cleavage and RIPK1 auto-phosphorylation than in WT cells (Fig. 4h and i). In addition, expression of constitutive activated MK2 in RIPK1 Y383F cells could alleviate but not fully inhibit RIPK1 kinase activity and TNF-induced cell death (Fig. 4j and k). These results suggest that MK2 deficiency is not the only cause of RIPK1 kinase hyper-activation caused by Y383F mutation. Since previous studies have demonstrated that RIPK1 phosphorylation mediated by IKK and TBK1 could directly suppress RIPK1 kinase activity, suggesting that tyrosine phosphorylation of RIPK1 might also have a directly inhibitory role on RIPK1 kinase activity besides MK2. Thus, we have modified our conclusions on the role of MK2 in protection mediated by p-Y383 of RIPK1.

Figure 5:

d,e: abbreviations such as LK, LSK, HSC, GMP, MEP and CMP should be explained

The authors should provide information on a possible phenotype in other organs of *Ripk1*^{Y383F}/*Y383F* mice such as the skin and lung that are affected in models of deregulated RIPK1 signalling such as *Sharpincpdm/cpdm* mice

RESPONSE: Thank you for pointing out our carelessness. As suggested, we have added the descriptions on LK, LSK, HSC, GMP, MEP and CMP in the manuscript (Page 13, line 238-247). In addition, we also did H&E examination and found no obvious inflammatory phenotype in skin, lung and kidney in *Ripk1*^{Y383F/Y383F} mice (Supplementary Fig. 5b).

Figure 6:

To show the dependency of systemic inflammation and emergency hematopoiesis in *Ripk1*^{Y383F}/*Y383F* mice on RIPK1 kinase activity, the authors could consider inhibiting RIPK1 kinase activity in these mice, for example by treatment with a small molecule inhibitor such as Nec1s.

RESPONSE: Thank you for your thoughtful comments. As suggested, we treated WT and Y383F mice with Nec1 inhibitors and found inhibition of RIPK1 kinase could indeed ameliorate the inflammation and emergency hematopoiesis in *Ripk1*^{Y383F/Y383F} mice (Fig. 6e and Supplementary Fig. 6a, b). However, due to the limitation of absorption and duration, RIPK1 inhibitors could not fully prevent the inflammation in *Ripk1*^{Y383F/Y383F} mice. In general, RIPK1 kinase hyperactivation is major cause of systemic inflammation and emergency hematopoiesis in *Ripk1*^{Y383F/Y383F} mice.

In general, the manuscript would benefit from an English language spelling and grammar check.

RESPONSE: Thank you for your thoughtful comment. As suggested, we have polished our manuscript both on the spelling and grammar.

Reviewer #2 (Remarks to the Author):

The authors investigated the role of tyrosine phosphorylation of RIPK1 in cell death and

inflammation, using MEF cells or mice with Y383 mutation. They provide evidence that non-receptor tyrosine kinases JAK1 and SRC bind with RIPK1 and phosphorylate RIPK1 at Y384 when overexpression on 293T cells. MEFs with RIPK1 Y383F blocks the MK2 phosphorylation, but promotes the RIPK1 activation and cell death including apoptosis and necroptosis under different treatment. Mice with Y383F mutation display systemic inflammation and emergency hematopoiesis which the authors suggest that can inhibited by deleting either TNFR1 or both RIPK3 and Caspase-8.

Regulation of RIPK1 by tyrosine phosphorylation is an area that has not been explored and thus has the potential to be interesting. However, the evidence provided is far from sufficient to demonstrate convincingly the presence of Y383(4) phosphorylation on RIPK1, the signaling mechanism that mediates this phosphorylation or its functional importance.

RESPONSE: We appreciate your positive comments on our study and pointing out some essential issues to improve our manuscript.

Main issue:

1) The authors started to investigate the role of JAK1 and SRC in the regulation of RIPK1 activity but somehow ended up on TNF which lacks logic as IFN is supposed to be involved in activating JAK and SRC, not TNF.

RESPONSE: Thank you for your thoughtful comment. The activation of JAK-STAT and Src kinase under TNF stimulation has been previously reported (PMID: 9510175; PMID: 11832448). Although IFN could also involved in activating JAK and Src kinase, we did not find significant alteration of IFN-mediated JAK-STAT activation by RIPK1 Y383F mutation (See attached data shown below).

2) The authors found that JAK1/ SRC bind with RIPK1 and phosphorylate RIPK1 at Y384 when overexpression on 293T cells. However, no result is provided for the binding or phosphorylation of endogenous RIPK1 by JAK1/SRC.

RESPONSE: Thank you for your comment. As suggested, we examined the interaction between endogenous JAK1/Src and RIPK1 in WT and Y383F primary BMDMs in response to TNF. Interestingly, we found JAK1 but not Src could interact with RIPK1 in TNFR1 complex1 with

TNF stimulation (Fig. 1d, e). However, Src kinase could interact with RIPK1 in complex II with TNF and BV-6 stimulation (Fig. 1f). These results indicate that JAK1 and Src indeed could bind to RIPK1 in different complexes to regulate TNFR1 signaling. In addition, we treated primary WT BMDMs with TNF or TNF-BV-6, and found RIPK1 could be phosphorylated in tyrosine residues. Consistently, pretreat with Jak1 or Src inhibitor could significantly block tyrosine phosphorylation on RIPK1 (Fig. 1i), suggesting that phosphorylation on RIPK1 depends on JAK1 or Src.

3) The data on MEFs that almost focus on cell death cannot fully support the inflammation result *in vivo*.

RESPONSE: Thank you for your comment. We did qPCR and ELISA analysis for NF- κ B targeting genes and found Y383F mutation of RIPK1 does not significantly affect induction of NF- κ B-targeted genes in primary BMDMs (Fig. 4d, e). However, our genetic data show that the inflammation could be fully rescued by deletion of Caspase8 and RIPK3, indicating that cell death is the major cause of the inflammation in *Ripk1*^{Y383F/Y383F} mice *in vivo* (Fig. 6f-i).

4)Y383F mutant promotes RIPK1 activation in complex-II but not affect the recruitment of RIPK1 in complex-I and the NF κ B activation. Whether knockout of JAK1/ SRC affect the complex-I or complex-II similarly with Y383F mutant, whether JAK1/ SRC be recruited into complex-I or complex-II required investigation.

RESPONSE: Thank you for your comment. As suggested, we examined the interaction between endogenous JAK1/Src and RIPK1 in WT and Y383F primary BMDMs in response to TNF. Interestingly, we found JAK1 but not Src could interact with RIPK1 in TNFR1 complex1 with TNF stimulation (Fig. 1d, e). However, Src kinase could interact with RIPK1 in complex II with TNF and BV-6 stimulation (Fig. 1f). In addition, since we have found JAK1 and Src could bind with RIPK1 and regulate its activity in TNFR1 complex I and complex II, respectively, we generated JAK1- and Src-deficient MEF cells to explore that function on complex formation. Consistently, JAK1 deficiency could largely block MK2 recruitment in complex1 but has no much influence on RIPK1 recruitment and ubiquitination, and Src deficiency could impair MK2 recruitment and promotes RIPK1 activation in complex II (Supplementary Fig. 4c, d).

5)Y383F mutant block the recruitment and phosphorylation of MK2, the loss function of p-MK2 is major effect by the Y383F mutant. It is reported that MK2 directly phosphorylates RIPK1 S321/336 in the cytosol but not in TNFR1 complex I (PMID: 28920952). As MK2 phosphorylates RIPK1 is directly, the mechanism how Y383F affect the MK2 activation still unclear.

RESPONSE: Thank you for your comment. Previous work indeed demonstrated MK2-mediated phosphorylation on RIPK1 is in cytosol, however, the recruitment and activation in TNFR1 complexes is not clearly answered. In this work, we did detect the recruitment and activation of MK2 in both TNFR1 complex1 and complex2 (Fig. 4f, g). Our previous published work also detected MK2 recruitment to TNFR1 complex1 in WT cells, which is significantly decreased in RIPK1 K63 ubiquitination defective K376R mutant cells (PMID: 31519887). These results suggest that MK2 could potentially function in TNFR1 complex to regulate cell death signaling.

In addition, the MK2 phosphorylation is also defective in Y383F mutation cells, suggesting that RIPK1 tyrosine phosphorylation is essential for MK2 activation and limiting RIPK1 kinase activity. However, we did not know the exact mechanism how tyrosine phosphorylation of RIPK1 leads to MK2 activation so far, which will be a topic for our future study. Phosphorylated tyrosine residue of substrates is well known for binding with SH2 domain containing proteins to activate downstream signaling. However, structural prediction in database suggests that MK2 is unlike to have SH2 domain for phospho-tyrosine residue binding. These suggest that other SH2 domain containing proteins might serve as an adaptor to bridge MK2 binding with tyrosine phosphorylated RIPK1. Thus, we added this discussion in the revised manuscript (Page 19, Line 362-372).

6) Y383F mutation induced systemic inflammation and emergency hematopoiesis in vivo. It is important to investigate if RIPK1 is activated in Y383F mutation mice, and whether the inflammation induced by Y383F mutation in mice is dependent on RIPK1 activation or could be inhibited by RIPK1 inhibitor.

RESPONSE: Thank you for your comments. As suggested, we treated WT and Y383F mice with Nec1 inhibitors and found inhibition of RIPK1 kinase could indeed ameliorate the inflammation and emergency hematopoiesis in *Ripk1*^{Y383F/Y383F} mice (Fig. 6e and Supplementary Fig. 6a, b). However, due to the limitation of absorption and duration, RIPK1 inhibitors could not fully prevent the inflammation in *Ripk1*^{Y383F/Y383F} mice. In general, RIPK1 kinase hyperactivation is major cause of systemic inflammation and emergency hematopoiesis in *Ripk1*^{Y383F/Y383F} mice.

(7) It is also important to examine the inflammation level on immune cells, like BMDM or PMBC cells by Y383/384 phosphorylation.

RESPONSE: Thank you for your comment. We did qPCR and ELISA analysis for NF-κB targeting genes and found Y383F mutation of RIPK1 does not significantly affect induction of NF-κB target genes in primary BMDMs (Fig. 4d, e).

(8) Certain key experimental data are missing. E.g Figure 6 is about comparing *Ripk1*-Y383F mutation in *Tnfr1*^{-/-} or *ripk3*^{-/-}*casp8*^{-/-}, but none of the comparing data included *Ripk1*-Y383F alone in Figure 6!

RESPONSE: Thank you for your comment. As suggested, we added the WT and Y383F mice control to re-examine the contribution of TNF-induced cell death to the inflammation and hematopoietic disorders in *Ripk1*^{Y383F/Y383F} mice (Fig. 6a-d, 6f-i and supplementary Fig. 6c-e).

Reviewer #3 (Remarks to the Author):

The manuscript by Tu et al. proposes that phosphorylation of RIPK1 at Y383 by the kinases SRC or JAK1 is required to limit cell death and inflammation. The authors argue that this tyrosine phosphorylation enables MK2 binding to RIPK1 and subsequent MK2-mediated phosphorylation to further inhibit the activity of RIPK1. The authors also present a knock-in model which shows

splenomegaly and emergency haematopoiesis, implying that this checkpoint on RIPK1 is relevant in vivo to prevent overt inflammatory phenotypes.

Despite these positive aspects, the data presented is often confusing and, to a major extent, relies on overexpression studies in HEK293 cells. Moreover, the conclusions drawn from the data obtained seem to be an oversimplification of the actual mechanisms at play. Therefore, the study would immensely benefit not only of a conceptual revisiting but also of additional endogenous characterization of crucial interactions. The in vivo phenotype of the new knock-in mutant seems clear, yet given the proposed mechanism, it is quite surprising that it is so different from those of MK2 knock-out and RIPK1 S321A knock-in mice. Moreover, the rescue with the RIPK3-Casp8 double knock-out raises further concerns that need to be clarified.

RESPONSE: We appreciate your positive comments on our study and pointing out some essential issues to improve our manuscript.

Major points:

Figure 1 and 2

1. While it is understandable that the early validation of the mass spec data and study of the domain involved in the binding is performed by overexpression studies, endogenous interactions must be shown for novel interactors. This is the expected standard in the field.

- Does endogenous RIPK1 bind to endogenous SRC and/or JAK1 and under which conditions, i.e. following which stimulus can these interactions be detected?

RESPONSE: Thank you for your comment. As suggested, we examined the interaction between endogenous JAK1/Src and RIPK1 in WT and Y383F primary BMDMs in response to TNF. Interestingly, we found JAK1 but not Src could interact with RIPK1 in TNFR1 complex 1 with TNF stimulation (Fig. 1d, e). However, Src kinase could interact with RIPK1 in complex II with TNF and BV-6 stimulation (Fig. 1f). These data indicate that endogenous JAK1 and Src could bind with RIPK1 in different complexes to regulate TNFR1 signaling.

- Is the binding of SRC and that of JAK1 to RIPK1 mutually exclusive or do these three proteins form a trimeric complex?

RESPONSE: Thank you for your comment. We co-expressed GFP-Src, Flag-RIPK1 and HA-JAK1 in 293T cells, however, we did not observe Src could interact with JAK1 with or without RIPK1 existence (See attached data shown below). These suggest that these three proteins do not form a trimeric complex.

2. The fact that the so called “ID” domain containing the RHIM domain is the domain required for the binding of RIPK1 to JAK1 is concerning. These constructs could bind to the full length RIPK1 that is (although at very low levels) still expressed in HEK293t via the reciprocal RHIM domain. The author needs to repeat these domain-mapping studies in HEK293 cells in which endogenous RIPK1 has been knocked out and cells have subsequently been reconstituted with the individual tagged constructs.

RESPONSE: Thank you for your comment. As suggested, we co-expressed Flag-RIPK1 and HA-JAK1 in RIPK1-deficient 293T cells and found RIPK1 still has a strong interaction with JAK1. In addition, we also found that the ID domain of RIPK1 could interact with JAK1 in the absence of endogenous RIPK1 in 293T cells (See attached data shown below). These suggest the ID domain of RIPK1 is indeed responsible for the interaction with JAK1.

3. Figure 1D is redundant. The fact that overexpressed proteins co-localize does not add anything to this manuscript. Also, this panel only shows three cells which is far too low a number to be of any significance.

RESPONSE: Thank you for your comment. As suggested, we deleted this data in revised manuscript.

Figure 3

4. The authors show that treatment with JAKi or SRCi enhances TNF sensitivity. Similar sensitivity is also detected in the RIPK1 Y383F knock-in cells treated with TNF. The amount of cell death detected is very low and goes from 10% to a little more than 15% when cells are co-treated with the inhibitors and in the case of the RIPK1 Y383F knock-in cells merely reaching a maximum 22% of cell death.

- How are these differences biologically significant? They may well be statistically significantly different but how can such low differences be sufficiently high to be biologically meaningful. The titration of higher concentrations of TNF and perhaps the additional co treatment with Smac

mimetic drugs is highly recommended to further investigate this point.

RESPONSE: Thank you for raising a critical issue. We repeated this cell death assay in primary BMDMs and show similar results that JAK and SRC inhibition or RIPK1 Y383F mutation shows relatively low TNF sensitivity (Fig. 3a). For the biological significance, we observed a relatively mild inflammatory phenotype in RIPK1 Y383F mice, which could be rescued by TNFR1 or Casp8/RIPK3 deficiency (Fig. 6a-d, 6f-i). These suggested the low level of TNF induced cell death, which triggered mild inflammation in Y383F mice. Moreover, we also used Smac mimetic drug BV-6 to accelerate TNF induced cell death and found that JAK and SRC inhibition or RIPK1 Y383F mutation indeed could promotes TNF-induced cell death in both MEFs and BMDMs (Fig. 3d and Supplementary Fig. 3a).

- The authors need to try both inhibitors in combination and, most importantly, they also need to treat the Y383F cells with the JAKi and/or SRCi.

RESPONSE: Thank you for your comment. As suggested, we found primary WT BMDMs treated with JAK and SRC inhibitor at the same time indeed has an enhanced cell death compared to treatment of JAK or SRC inhibitor alone (Fig. 3a). Interestingly, treatment of JAK+SRC inhibitors had a similar effect as Y383F mutation (Fig. 3b). These results suggest that JAK and SRC has a non-redundant function in protection from TNF-induced cell death and this effect is through phosphorylation of Y383.

Figure 4

5. The result shown in Figure 4A contradicts the main message of this manuscript. While lane 2 is clearly darker, it is impossible to detect any difference for the RIPK1 band. Also, the blots are heavily overexposed and it is therefore impossible to judge the quality of the immunoprecipitation. This needs to be repeated and higher quality data need to be provided.

RESPONSE: Thank you for your comment. As suggested, we repeated this experiment in primary BMDMs from WT and Y383F mice, and found that Y383F mutation greatly impairs tyrosine phosphorylation of RIPK1 in response to TNF α treatment (Fig. 4a).

6. Figure 4E needs to be done under endogenous conditions. The authors have the knock-in cells and the conditions under which MK2 and RIPK1 are bound have been extensively published by three different groups. There is no reason why this data should be presented by utilizing an overexpression system.

RESPONSE: Thank you for your comment. As suggested, we used WT and Y383F MEFs to perform the interaction between endogenous RIPK1 and MK2, and found Y383F mutation indeed prevents RIPK1 from binding with MK2 under TNF/BV-6 stimulation (Fig. 4g).

7. Figure 4D, F, G show blots for pMK2. Why is total MK2 not shown (see point above)? Also, in all these three panels loss of pMK2 at the complex coincides with the overall loss of phosphorylation in the lysates. This is rather unexpected and would suggest that the Y383F

mutation is somehow impacting the activation of MK2 which, canonically, is known to be independent of RIPK1 kinase activity. This needs to be further investigated because it points towards a direct effect of the mutant on MK2 activation and/or stability which would mean that the mechanism would be quite different from the one currently proposed by the authors the manuscript.

RESPONSE: Thank you for your comment. We also detected MK2 recruitment to TNFR1 complex1 and complex2, and found Y383F mutation did not affect MK2 stability but impairs its recruitment to these complexes (Fig. 4f, g). In addition, the MK2 phosphorylation is also defective in Y383F mutation cells, suggesting that RIPK1 tyrosine phosphorylation is essential for MK2 activation and limiting RIPK1 kinase activity. However, we did not know the exact mechanism how tyrosine phosphorylation of RIPK1 leads to MK2 activation so far, which will be a topic for our future study. Phosphorylated tyrosine residue of substrates is well known for binding with SH2 domain containing proteins to activate downstream signaling. However, structural prediction in database suggests that MK2 is unlike to have SH2 domain for phospho-tyrosine residue binding. These suggest that other SH2 domain containing proteins might serve as an adaptor to bridge MK2 binding with tyrosine phosphorylated RIPK1. Thus, we added this discussion in the revised manuscript (Page 19, Line 362-372).

Figure4G-H:

8. Are the Sh-RNA utilised pools? How many were tested? More than one should be included to avoid potential off-target effects.

RESPONSE: Thank you for your comment. Our shRNA for MK2 indeed utilized pools, and as suggested, we generated two MK2-deficient cell lines with good Knockdown efficiency to repeat these experiments (Fig. 4h, i).

- In figure 4G it is clear that the ShMK2 alone has an effect on cleavage of Casp3 as compared to ShCtr. Interestingly, while the cleavage of Casp3 is enhanced in the Y383F cells where the levels of MK2 have been depleted these are considerably lower than the levels observed in control cells. These results raise numerous questions. If the effect of the Y383 phosphorylation is via MK2, why is it that Y383F shows more cleavage of Casp3 in the absence of MK2? Why are they lower than in control cells? The mutant should sensitize to TNF. This seems to contradict the results of Fig 3E and Fig 4H where the knockdown of MK2 shows no further sensitization to the knock-in mutation. Do the authors envisage more than one modality of cell death at play which does not involve only cleavage of Casp3? This needs to be revisited. As it stands, the mechanism is.

RESPONSE: Thank you for your comment. We generated two MK2-deficient cell lines with good Knockdown efficiency to repeat these experiments, and found that MK2 deficiency in Y383F cell still has higher caspase-3 cleavage and RIPK1 auto-phosphorylation than in WT cells (Fig. 4h and i). In addition, expression of constitutive activated MK2 in RIPK1 Y383F cells could alleviate but not fully inhibit RIPK1 kinase activity and TNF-induced cell death (Fig. 4j and k). These results suggest that MK2 deficiency is not the only cause of RIPK1 kinase hyper-activation caused by

Y383F mutation. Since previous studies have demonstrated that RIPK1 phosphorylation mediated by IKK and TBK1 could directly suppress RIPK1 kinase activity, suggesting that tyrosine phosphorylation of RIPK1 might also have a directly inhibitory role on RIPK1 kinase activity besides MK2. Thus, we have modified our conclusions on the role of MK2 in protection mediated by p-Y383 of RIPK1.

Figure 6

9. Figure 6 should also include the data for the knock-in mice. It is impossible to draw any conclusion otherwise. More importantly, how do the authors justify the same number of neutrophils in the TNFR1 knock-out mouse as in the RIPK3-Casp8 double knock-out mouse (0.77 Vs 0.71, Figure 6B, F). Moreover, and most importantly, the RIPK3-Casp8 knock-out mouse suffers from lymphadenopathy and splenomegaly (akin Fas-lpr or FasL-gld mice). This is clearly RIPK1-independent as RIPK3- Casp8-RIPK1 triple knock-out mice also present with this phenotype. How can the levels of hematopoietic cells be normal in these mice? The 5 genotypes (including the knock-in) need to be compared side by side. This includes the cell sorting with same gating and the plotting and the percentage normalization on the same graphs. This current representation of the data does not account for the underlying phenotype of the double knock-outs and is therefore misleading. The conclusion that RIPK1 Y383F-induced inflammation and cell death is rescued by co-deletion of Casp8 and RIPK3 cannot be drawn from the presented data and must be properly analyzed.

RESPONSE: Thank you for your comment. we felt so sorry for lacking the knock-in mice as control and misleading you on the TNFR1 and RIPK3-Casp8 knock-out mouse. For the concerns on lymphadenopathy and splenomegaly of RIPK3-Casp8 knock-out mouse, we analyzed the inflammatory phenotype of RIPK3-Casp8 knock-out mouse at 6-8 week, which is too early to develop LPR disease. Thus, the FACS analysis showed the neutrophil and other hematopoietic cell percentage is similar as TNFR1 KO mice. Due to lack of RIPK1-RIPK3-Caspase-8 triple Knock-out mice strain in our hands, we compared WT, *Ripk1*^{Y383F/Y383F}, *casp8*^{-/-}*Ripk3*^{-/-} and *Ripk1*^{Y383F/Y383F} *Casp8*^{-/-}*Ripk3*^{-/-} mice side by side at a young age at 8 weeks and old age at 16 weeks. Similar with our previous data, *casp8*^{-/-}*Ripk3*^{-/-} and *Ripk1*^{Y383F/Y383F} *Casp8*^{-/-}*Ripk3*^{-/-} mice at 8 weeks did not develop lymphadenopathy and splenomegaly and has no obvious inflammation and hematopoietic disorders compared to *Ripk1*^{Y383F/Y383F} mice (Fig. 6f-i). However, both *casp8*^{-/-}*Ripk3*^{-/-} and *Ripk1*^{Y383F/Y383F} *Casp8*^{-/-}*Ripk3*^{-/-} mice at age of 16 weeks developed severe lymphadenopathy and splenomegaly which mainly due to significantly enhanced CD3+B220+ cell populations. Moreover, compared to *Ripk1*^{Y383F/Y383F} mice, the inflammatory neutrophils infiltration in *Ripk1*^{Y383F/Y383F} *Casp8*^{-/-}*Ripk3*^{-/-} mice is greatly decreased to a similar level as in *casp8*^{-/-}*Ripk3*^{-/-} mice (Supplementary Fig. 6c-e). These suggest that the inflammation in *Ripk1*^{Y383F/Y383F} mice is caused by enhanced cell death.

REVIEWER COMMENTS

Reviewer #1 (Remarks to the Author):

The authors have largely addressed my concerns. Particularly the new experiments in primary BMDMs provide a more clear picture and support the key conclusions of the authors. Please see below some last suggestions which should be considered before publication.

The data provided by the authors suggest that the effect of Tyr phosphorylation is only partly mediated via the recruitment of MK2. However, both in the abstract and in the main text, the description of the role of MK2 gives the impression that this is the only mechanism. For example, the abstract writes: Lines 21-22: "Mechanistically, Ripk1Y383F/Y383F mutation blocks recruitment and activation of MK2 and promotes RIPK1 kinase activation" and Lines 79-80: "we provide the precise molecular and genetic mechanism by which tyrosine phosphorylation of RIPK1 on Y383 orchestrates RIPK1-dependent cell death and further regulates inflammation". It is clear that MK2 contributes to the Tyr phosphorylation effect but other, yet unknown mechanisms seem to be at play as well. This needs to be clearly stated throughout the paper.

In Figure 3D, it is important to show if Nec1 (ideally Nec1s) prevents TBZ-induced necroptosis in addition to inhibiting TB-induced apoptosis. TBZ+Nec1 treatment is obviously missing from this figure panel particularly because this is included in the relevant western blot in 3F.

Line 181: MEF experiments cannot be described as *in vivo*, this would refer to live animals.

Line 192: All that figures 4d,e show is TNF expression and should be described as such. One can then discuss that this supports that TNF-induced inflammatory gene expression is not suppressed, but it should be clear that this is based only on measuring TNF expression.

Fig 5a: please indicate the sex of the mice as male mice are usually heavier than female so it is important to have this comparison performed on sex-matched animals.

Please make sure all information is provided for the reader to understand the results, e.g. in Fig 3D the time point of analysis is not described.

Reviewer #3 (Remarks to the Author):

The authors have addressed all previously raised concerns. The authors took into account all of suggestions I made and modified their interpretation of the data and the overall mechanism interpretation according not only to the experiments that I have indicated in the revision round (and now added to this manuscript) but also the overall interpretation discussed in the referee report. While the cell death assay still show only a minor *in vitro* phenotype, the data *in vivo* are clear and the biochemistry has largely improved. The repeated blots now contain proper controls and have been repeated in appropriate knock out cells to provide a clearer picture of the different complexes.

The only remaining minor points are the following:

1. The authors need to thoroughly go through the spelling of the manuscript, provide, in the figure legends and in the text, accurate concentrations of the different drugs used, their exact name and specs, and precise time points of the experiments.
2. The authors should also implement accurate labellings of all the western blots. This is especially the case for phospho-proteins: it would be advisable to specify the phospho-residue recognized by the antibody and make sure that there is a uniform labelling throughout.
3. The authors should add the blots used only for the rebuttal to the supplementary data and mention these in the manuscript, as these would answer questions that other readers might also have.

Reviewer #4 (Remarks to the Author):

In the study entitled "Tyrosine phosphorylation regulates RIPK1 activity to limit cell death and inflammation", Tu et al. describe a novel phosphorylation event that regulates RIPK1 function in response to TNF. This is a tyrosine phosphorylation that is mediated by JAK and SRC kinases and can function in concert with MK2 regulation on RIPK1. The authors examine this mechanism both in the context of in vitro experiments as well as by generating a knock-in mouse model which has a relatively mild phenotype. In these experiments the authors show that this tyrosine phosphorylation event can regulate cell death and suggest that JAK/SRC in the TNFR1 complexes can regulate RIPK1 function. The authors have addressed most of reviewer 2's comments; some specific comments are described below:

Re issue #1: The authors show that the Y383F mutation does not alter IFN-dependent Jak/stat phosphorylation and signaling, however they do not show any data that there are alterations of this signaling pathway in response to TNF by WB. It is potentially implied based on the effect of the Jak/Src kinase inhibitors and shRNA experiments is this signaling specific to TNF or other death receptors or TLRs?

Re #2 and #4: While the authors assume that complex 2 is formed after >1 h following stimulation they don't show the presence and/or of caspase 8 and other components.

#3 While the data of inflammatory response in MEFS in response to TNF is clear it may have been useful to look at the inflammatory response in the presence of an IAP inhibitor to confirm that RIPK1 dependent inflammation is not altered in this condition.

#5 IP of pMK2 vs. total MK2 looks a little strange, in terms of the complete absence of the protein at 15 min .

#6 The authors show some effect of Nec1s on the hematopoietic phenotype however it would have been nice to include the alteration of pRIPK1 in these cells to show that the cellular and tissue changes correlate with alterations in kinase activity.

#7 The authors could have done an ELISA in the animals to show that cellular changes seen in the mice also correlate with robust changes in cytokine/chemokine levels that are typically altered upon RIPK1 activation

Based on reviewers' comments and critiques, we have corrected our manuscript and performed a series of experiments as shown in Fig. 3d, 3g, 4d and supplementary Fig. 1c, 1d, 2a, 2b, 4d, 4e, 4f, 4g, 5b. In addition, we also provide the following point-to-point response to reviewers' questions.

REVIEWER COMMENTS

Reviewer #1 (Remarks to the Author):

The authors have largely addressed my concerns. Particularly, the new experiments in primary BMDMs provide a more clear picture and support the key conclusions of the authors. Please see below some last suggestions which should be considered before publication.

The data provided by the authors suggest that the effect of Tyr phosphorylation is only partly mediated via the recruitment of MK2. However, both in the abstract and in the main text, the description of the role of MK2 gives the impression that this is the only mechanism. For example, the abstract writes: Lines 21-22: "Mechanistically, Ripk1Y383F/Y383F mutation blocks recruitment and activation of MK2 and promotes RIPK1 kinase activation" and Lines 79-80: "we provide the precise molecular and genetic mechanism by which tyrosine phosphorylation of RIPK1 on Y383 orchestrates RIPK1-dependent cell death and further regulates inflammation". It is clear that MK2 contributes to the Tyr phosphorylation effect but other, yet unknown mechanisms seem to be at play as well. This needs to be clearly stated throughout the paper.

RESPONSE: We appreciate your positive comments on our study and pointing out some critical issues to improve our manuscript. As suggested, we corrected our statement of the molecular mechanism how MK2 involved in the regulation of RIPK1 kinase activity via tyrosine phosphorylation both in the abstract and main text (Page 2, Line 21-23; Page 5, Line 77-81). In addition, we also added discussion of exclusive mechanisms that might underpin the suppressive function of tyrosine phosphorylation on RIPK1 kinase activity (Page 18, Line 346-355).

In Figure 3D, it is important to show if Nec1 (ideally Nec1s) prevents TBZ-induced necroptosis in addition to inhibiting TB-induced apoptosis. TBZ+Nec1 treatment is obviously missing from this figure panel particularly because this is included in the relevant western blot in 3F.

RESPONSE: Thank you for your thoughtful suggestions. As suggested, we added TBZ+Nec1 treatment to repeat this experiment and found that Nec-1s could also prevent TBZ-induced necroptosis in RIPK1 Y383F BMDMs (Fig. 3d).

Line 181: MEF experiments cannot be described as in vivo, this would refer to live

animals.

RESPONSE: Thank you for your thoughtful comments. We have corrected this description in the manuscript (Page 10, Line185-186).

Line 192: All that figures 4d,e show is TNF expression and should be described as such. One can then discuss that this supports that TNF-induced inflammatory gene expression is not suppressed, but it should be clear that this is based only on measuring TNF expression.

RESPONSE: Thank you for your thoughtful comments. We felt sorry for misleading you on our results and description. To avoid this, we treated BMDMs with TNF and found the transcriptional level of inflammatory genes such as IL1 β and CXCL10 did not have significant alteration in Y383F BMDMs compared to WT controls (Fig. 4d). These further suggest that *Ripk1*^{Y383F/Y383F} mutation has no effect on TNF α -induced inflammatory signaling activation. In addition, we also modified our description on these results in the manuscript (Page 11, Line 196-200).

Fig 5a: please indicate the sex of the mice as male mice are usually heavier than female so it is important to have this comparison performed on sex-matched animals.

RESPONSE: Thank you for your thoughtful comments. Actually, we all used sex-matched animals to monitor the body weight. As suggested, we have added the description of sex of the mice in the manuscript (Page 30, Line 666-667).

Please make sure all information is provided for the reader to understand the results, e.g. in Fig 3D the time point of analysis is not described.

RESPONSE: Thank you for your thoughtful comments. As suggested, we carefully examined our manuscript and added those descriptions.

Reviewer #3 (Remarks to the Author):

The authors have addressed all previously raised concerns. The authors took into account all of suggestions I made and modified their interpretation of the data and the overall mechanism interpretation according not only to the experiments that I have indicated in the revision round (and now added to this manuscript) but also the overall interpretation discussed in the referee report. While the cell death assay still show only a minor in vitro phenotype, the data in vivo are clear and the biochemistry has largely improved. The repeated blots now contain proper controls and have been repeated in appropriate knock out cells to provide a clearer picture of the different complexes.

RESPONSE: We appreciate your positive comments on our study and pointing out some critical issues to improve our manuscript.

The only remaining minor points are the following:

1. The authors need to thoroughly go through the spelling of the manuscript, provide, in the figure legends and in the text, accurate concentrations of the different drugs used, their exact name and specs, and precise time points of the experiments.

RESPONSE: Thank you for your thoughtful comments. As suggested, we carefully examined our manuscript and added those descriptions.

2. The authors should also implement accurate labelling of all the western blots. This is especially the case for phospho-proteins: it would be advisable to specify the phospho-residue recognized by the antibody and make sure that there is a uniform labelling throughout.

RESPONSE: Thank you for your thoughtful comments. As suggested, we carefully examined our figures and changed the labelling.

3. The authors should add the blots used only for the rebuttal to the supplementary data and mention these in the manuscript, as these would answer questions that other readers might also have.

RESPONSE: Thank you for your thoughtful comments. As suggested, we added rebuttal blots into the supplementary data and mentioned in the manuscript.

Reviewer #4 (Remarks to the Author):

In the study entitled "Tyrosine phosphorylation regulates RIPK1 activity to limit cell death and inflammation", Tu et al. describe a novel phosphorylation event that regulates RIPK1 function in response to TNF. This is a tyrosine phosphorylation that is mediated by JAK and SRC kinases and can function in concert with MK2 regulation on RIPK1. The authors examine this mechanism both in the context of in vitro experiments as well as by generating a knock-in mouse model which has a relatively mild phenotype. In these experiments the authors show that this tyrosine phosphorylation event can regulate cell death and suggest that JAK/SRC in the TNFR1 complexes can regulate RIPK1 function. The authors have addressed most of reviewer 2's comments; some specific comments are described below:

RESPONSE: We appreciate your positive comments on our study and pointing out some critical issues to improve our manuscript.

Re issue #1: The authors show that the Y383F mutation does not alter IFN-dependent Jak/stat phosphorylation and signaling, however they do not show any data that there are alterations of this signaling pathway in response to TNF by WB. It is potentially

implied based on the effect of the Jak/Src kinase inhibitors and shRNA experiments is this signaling specific to TNF or other death receptors or TLRs?

RESPONSE: Thank you for your thoughtful comments. Our results have shown JAK1 and its kinase activity is essential to phosphorylate RIPK1 at Y383 and suppress RIPK1-dependent cell death. However, we did not investigate whether RIPK1 Y383F mutation could affect TNF-induced JAK1/STAT activation. As suggested, we treated WT and RIPK1 Y383F mutant BMDMs with TNF or LPS. However, we did not observe significant difference of JAK1/STAT1 activation between WT and Y383F group (Supplementary Fig. 4d-f). These results suggest that JAK1 is activated upstream of RIPK1 in TNF signaling and phosphorylate RIPK1 without affecting downstream activation of STAT1. Since RIPK1 could also participate in TLR-mediated NF- κ B and cell death signaling, we also test the role of JAK1-mediated phosphorylation of RIPK1 in TLR signaling. Interestingly, we also found JAK1 inhibition or Y383F mutation could also enhance LPS+BV6-induced apoptosis in BMDMs (See attached data shown below), suggesting that JAK1-mediated phosphorylation of RIPK1 involved in both TNF and TLR-mediated cell death regulation.

Re #2 and #4: While the authors assume that complex 2 is formed after >1 h following stimulation they don't show the presence and/or of caspase 8 and other components.

RESPONSE: Thank you for your thoughtful comments. We repeated this experiment and examined caspase-8 recruitment in complex 2. As complex 2 formation is under apoptotic condition, we found more cleaved caspase-8 interacted with RIPK1, which is consistent with more RIPK1-associated FADD, cleaved cFLIP and auto-phosphorylated RIPK1 in RIPK1 Y383F mutant cells (Fig. 3g). Thus, these results suggested that RIPK1 Y383F mutation promotes complex 2 formation under TNF stimulation.

#3 While the data of inflammatory response in MEFS in response to TNF is clear it may have been useful to look at the inflammatory response in the presence of an IAP inhibitor to confirm that RIPK1 dependent inflammation is not altered in this condition.

RESPONSE: Thank you for your thoughtful comments. As suggested, we treated WT and RIPK1 Y383F BMDMs with or without IAP inhibitor under stimulation of TNF

and then used qPCR to detect inflammatory genes expression. Notably, we observed that pretreatment of IAP inhibitor BV6 could greatly enhance transcriptional level of TNF, IL1b and CXCL10 with or without TNF treatment (Supplementary Fig. 4g). However, we did not observe significant difference of these inflammatory gene expression between WT and Y383F BMDMs under TNF stimulation with or without BV6 treatment (Supplementary Fig. 4g). These results further suggested that the inflammation in *Ripk1*^{Y383F/Y383F} mice is caused by enhanced cell death-mediated inflammatory response but not by transcriptional upregulation of inflammatory signaling.

#5 IP of pMK2 vs. total MK2 looks a little strange, in terms of the complete absence of the protein at 15 min.

RESPONSE: Thank you for your thoughtful comments. We felt so sorry for misleading you with our data in Fig. 4f. The pattern of MK2 and p-MK2 recruitment in TNF-RSC seems to be not consistent. We repeated this experiment and found that the MK2 recruitment in TNF-RSC is actually detectable both in 5 and 15 min, which is consistent with p-MK2 (Fig. 4e). We carefully examined the raw data of original blots of MK2 but did not find any mistakes. It is possible that there might be some problem during the antibody incubation and western blotting exposure process of original blots. Thus, we changed this blot in Fig. 4f.

#6 The authors show some effect of Nec1s on the hematopoietic phenotype however it would have been nice to include the alteration of pRIPK1 in these cells to show that the cellular and tissue changes correlate with alterations in kinase activity.

RESPONSE: Thank you for your suggestions. We did show Nec1s could alleviate the hematopoietic disorders including enhanced GMP and decreased MEP population in Supplementary Fig. 6b. It is a good suggestion to detect p-RIPK1 in these hematopoietic cells. As you know, Western blotting using pRIPK1 antibody is the most efficient way to detect RIPK1 activation both in commercial and in our hands. However, the population of these hematopoietic progenitor cell in bone marrow is too small to harvest enough protein samples to detect p-RIPK1 alteration. Since our previous results in Figure 3 have shown a significantly enhanced p-RIPK1 in the primary bone marrow derived macrophages under TNF stimulation, the effect of nec1 on the hematopoietic phenotype in RIPK1 Y383F mice should be correlated with alteration in its kinase activity.

#7 The authors could have done an ELISA in the animals to show that cellular changes seen in the mice also correlate with robust changes in cytokine/chemokine levels that are typically altered upon RIPK1 activation;

RESPONSE: Thank you for your thoughtful comments. As suggested, we used ELISA to examine cytokine level in the serum and found that inflammatory cytokine IL1 β and

IL6 is also slightly enhanced in RIPK1 Y383F mice compared to WT mice (Supplementary Fig. 5b), which is correlated with cellular changes of inflammatory immune cells.

REVIEWERS' COMMENTS

Reviewer #1 (Remarks to the Author):

The authors have addressed all my concerns. I recommend publication of the manuscript.

Reviewer #3 (Remarks to the Author):

The authors have satisfactorily addressed my remaining concerns in the re-revised version of this manuscript.

Reviewer #4 (Remarks to the Author):

The authors have addressed all of my concerns

REVIEWERS' COMMENTS

Reviewer #1 (Remarks to the Author):

The authors have addressed all my concerns. I recommend publication of the manuscript.

Response: Thanks a lot.

Reviewer #3 (Remarks to the Author):

The authors have satisfactorily addressed my remaining concerns in the re-revised version of this manuscript.

Response: Thanks a lot.

Reviewer #4 (Remarks to the Author):

The authors have addressed all of my concerns

Response: Thanks a lot.